# Twin modelling reveals partly distinct genetic pathways to music enjoyment

Giacomo Bignardi [1,2] ✉, Laura W. Wesseldijk [3,4,5], Ernest Mas-Herrero [6,7,8], Robert J. Zatorre [9,10], Fredrik Ullén [3,5,14], Simon E. Fisher [1,11,14] & Miriam A. Mosing [3,5,12,13,14]

Humans engage with music for various reasons that range from emotional regulation and relaxation to social bonding. While there are large inter-individual differences in how much humans enjoy music, little is known about the origins of those differences. Here, we disentangle the genetic factors underlying such variation. We collect data on several facets of music reward sensitivity, as measured by the Barcelona Music Reward Questionnaire, plus music perceptual abilities and general reward sensitivity from a large sample of Swedish twins (N = 9169; 2305 complete pairs). We estimate that genetic effects contribute up to 54% of the variability in music reward sensitivity, with 70% of these effects being independent of music perceptual abilities and general reward sensitivity. Furthermore, multivariate analyses show that genetic and environmental influences on the different facets of music reward sensitivity are partly distinct, uncovering distinct pathways to music enjoyment and different patterns of genetic associations with objectively assessed music perceptual abilities. These results paint a complex picture in which partially distinct sources of variation contribute to different aspects of musical enjoyment.

Music can evoke intense pleasure and induce various emotions[1–3], leading individuals across different cultures[4] to actively seek out and engage with it. This human attraction to music has always been considered somewhat baffling[5] and mysterious[6], leading many to ask why music has such power over humans[7,8]. Oliver Sacks highlighted this conundrum in the opening of his beautifully written commentary, The Power of Music: "What an odd thing it is", he wrote " to see an entire species—billions of people—playing with listening to meaningless tonal patterns, occupied and preoccupied for much of their time by what they call 'music'"[8]. Despite the widespread power of music, however, not everyone responds to it to the same extent. Within human populations, there is indeed ample evidence that music-related cognition, from perceptual to affect-related processes, varies from one person to another[9–12].

Over the last decade, several studies have explored such differences between individuals in music-related traits and states, to better

[1]Language and Genetics Department, Max Planck Institute for Psycholinguistics, Nijmegen, the Netherlands. [2]Max Planck School of Cognition, Leipzig, Germany. [3]Department of Neuroscience, Karolinska Institutet, Stockholm, Sweden. [4]Department of Psychiatry, Amsterdam UMC, University of Amsterdam, Amsterdam, the Netherlands. [5]Department of Cognitive Neuropsychology, Max Planck Institute for Empirical Aesthetics, Frankfurt am Main, Germany. [6]Department of Cognition, Development and Educational Psychology, Universitat de Barcelona, Barcelona, Spain. [7]Institute of Neurosciences, Universitat de Barcelona, Barcelona, Spain. [8]Cognition and Brain Plasticity Group, Institut d'Investigació Biomèdica de Bellvitge (IDIBELL), Hospitalet de Llobregat, Barcelona, Spain. [9]Montreal Neurological Institute, McGill University, Montreal, QC, Canada. [10]International Laboratory for Brain, Music and Sound Research (BRAMS), Montreal, QC, Canada. [11]Donders Institute for Brain, Cognition and Behaviour, Radboud University, Nijmegen, The Netherlands. [12]Department of Medical Epidemiology and Biostatistics, Karolinska Institutet, Solna, Sweden. [13]Melbourne School of Psychological Sciences, Faculty of Medicine, Dentistry, and Health Sciences, University of Melbourne, Melbourne, VIC, Australia. [14]These authors jointly supervised this work Fredrik Ullén, Simon E. Fisher and Miriam A. Mosing. ✉e-mail: Giacomo.bignardi@mpi.nl

understand the basis of human musicality[13], broadly defined as the capacity to perceive, produce, and enjoy music. These studies show that individual differences in musicality correlate with neurobiological differences[14–16]. For example, the study of individuals with lifelong musical pitch deficits underscores the relevance of brain connectivity patterns in distributed neural networks for conscious perception of music[16]. Similarly, studies of differences in musical enjoyment highlight how interactions between cortical and subcortical brain regions support perceptual and affective processes that are fundamental for the experience of musical pleasure[14,15,17–19]. Moreover, recent studies have started to uncover the role of genetic factors in perceptual-motor processing of music[20] (e.g., the ability to synchronise with an external beat or recognise a melody) as well as in music production, such as levels of musical achievement[21,22] and engagement[23,24]. In general, these studies highlight complex gene-environment interplay[25,26] and the involvement of many DNA variants[20], each with a small effect (see ref. [27]).

Despite the many studies that have examined differences in music-related traits, still little is known about the genetic sources of differences in affective aspects of music processing and, in particular, the ability to enjoy music[28,29]. A better understanding of such genetic effects will elucidate how the ability to enjoy music is passed from one generation to the other, clarify the mechanisms linking genotypes, brains, and affect, and contribute to solving the conundrum of how "meaningless tonal patterns" can have such a powerful effect on humans.

Here, we study individual differences in musical enjoyment, focusing on music reward sensitivity, as measured by the Barcelona Music Reward Questionnaire (BMRQ)[11,15]. We used the BMRQ as it is a validated and reliable (e.g., one-year test-retest reliability of $r = 0.94$, see ref. [11]) instrument that provides a total score to assess how much individuals derive pleasure from music, as well as fine-grained characterisation of individual differences in emotion evocation (emotional reactions to music), mood regulation (the impact of music on individuals' moods), music seeking (the tendency to seek out new music), sensory motor (deriving pleasure from movements evoked by music), and social reward (deriving pleasure from social bonding through music) facets of music enjoyment[10]. Overall, the BMRQ is a well-established psychometric tool in the music science literature, showing robust associations with affective experiences[30–32], cognition[33–35], physiology[11], and neurobiology[14,15,36,37].

When studying sources of individual differences in music reward sensitivity, it is also crucial to consider variation in more general perceptual or hedonic processes, as the former can be merely a consequence of the latter (e.g., a specific lack of pleasure derived from music could be one aspect of a general lack of sensitivity to rewarding stimuli). As such, we complemented our study by including measures of music perceptual abilities and general reward sensitivity, and addressed the following three research questions:

1. To what extent are differences in music reward sensitivity (as measured by the BMRQ) explained by genetic variation?
2. To what extent do genetic effects influence music reward sensitivity above and beyond genetic effects shared with music perceptual abilities and general reward sensitivity?
3. To what extent are genetic effects shared between the different facets of music reward sensitivity?

To address these questions, we utilised a large sample of twins with available musicality data. We addressed the first question by estimating the heritability of music reward sensitivity using the Classical Twin Design (CTD)[38]. We addressed the second question by applying multivariate twin modelling to estimate the genetic overlap between music reward sensitivity (BMRQ), music perceptual ability based on a composite score of the melody, pitch, and rhythm scales of the Swedish Musical Discrimination Test (SMDT)[12], and general reward sensitivity, measured with the Behavioral Approach System Reward Responsiveness (BAS-RR) sub-scale[39], which has previously been

shown to correlate with the BMRQ[10,11]. The third question was assessed by testing if genetic effects are shared across facets of music reward sensitivity, consistent with a shared genetic factor of music enjoyment, or whether, alternatively, genetic influences are largely distinct for each facet. Additionally, we conducted two multivariate analyses at the facet level with music perceptual abilities or general reward sensitivity, respectively. All analyses were complemented by an investigation of environmental influences on music reward sensitivity and their overlap with environmental effects on music perceptual ability and general reward sensitivity.

## Results

### Sample and BMRQ descriptives
We utilised self-reported BMRQ data in a sample of 9169 monozygotic (MZ) and same-sex and opposite-sex dizygotic (DZ) Swedish twins, with a mean (M) age of 51 years (standard deviation (σ) = 8 years, range from 37 to 64 years; see Table 1 for sample size split by sex and zygosity; see "Methods" section for details on the cohort and zygosity identification). Since previous work in different populations used the BMRQ total score to measure music reward sensitivity—finding associations with psychological, physiological, and neurobiological variables—we began our analysis at the BMRQ total score level. A confirmatory factor model showed an acceptable fit in the Swedish sample for a model with a single latent music reward sensitivity factor capturing correlations between the five facets (CFI = 0.96, SRMR = 0.035). BMRQ total scores ranged from 20 to 100, with M = 71.20 and σ = 13.95. In line with previous studies, the BMRQ distribution was negatively skewed (skew = −0.58; i.e., a long tail of individuals with lower BMRQ total scores; see Supplementary Fig. 1).

### Genetic factors play a substantial role in music reward sensitivity
To estimate to what extent genetic effects (A: additive; D: dominance), the family environment shared between members of a family (C: common environment), and residual experiences unique to each individual (E: non-common environment, including measurement error) influence music reward sensitivity, we used Structural Equation Modelling (SEM), informed by the CTD. First, as a baseline for further model comparisons, we fitted a saturated model to individuals' BMRQ total scores (Fig. 1a; age and sex were accounted for). Assumptions of equality of means and variance across zygosities, twins within a pair, and sex were met (see Supplementary Table 3), except for the equality of means across sex: Consistent with previous literature[40], BMRQ scores were higher in women (M = 76.26) than in men (M = 71.20) (sex-constrained σ = 13.72; Cohens' d = 0.37, 95% CI [0.33, 0.41], $\chi^2(\Delta df = 30) = 300.54$, $p < 0.001$; note that throughout the study, we used an alpha of α = .007; see "Methods" section for justification). We, therefore, did not constrain means to be equal between sexes in subsequent models. Also consistent with previous results[10,41], age was negatively associated with overall BMRQ scores, although the effect was small, $B_{age} = −0.06$ (95% CI [−0.10,−0.02]; $\sigma_{age} = 57.90$ years, $\chi^2(\Delta df = 1) = 8.58$, $p = 0.003$), explaining only 0.1% of the total BMRQ variance. Since the skewness of BMRQ scores was below 2 (see ref. [42]), all SEM analyses used the full-information maximum likelihood estimator. Analyses using alternative estimators, robust to departures from multivariate normality, did not change the findings; the results of these analyses are provided in Supplementary Note 1.

By comparing within-pair MZ and DZ correlations of BMRQ scores, we estimated the heritability of music reward sensitivity, i.e., the proportion of phenotypic variance in this trait which is explained by genetic variation[38]. Twin correlations for music reward sensitivity were higher for MZ ($r_{MZ} = 0.55$, 95% CI [0.51, 0.59]) than DZ ($r_{DZ} = 0.24$, 95% CI [0.19, 0.29]) twins (Fig. 1b, see Supplementary Fig. 2). As the $r_{MZ}$ was more than twice the $r_{DZ}$, a model with additive and dominance genetic components (ADE) was fit (Fig. 1c). The ADE model reasonably

**Table 1 | Numbers of monozygotic (MZ) and dizygotic (DZ) twin pairs for each trait**

| Trait | Measure | Wave | MZ women *n* (*n* pairs) | MZ men *n* (*n* pairs) | DZ women *n* (*n* pairs) | DZ men | DZ os | Total twins |
|---|---|---|---|---|---|---|---|---|
| Music perceptual abilities[+] | Swedish Musical Discrimination Test (SMDT) | Humans Making Music 1 (2013) | 1012 (357) | 632 (200) | 705 (201) | 525 (128) | 1162 (280) | 4036 (716) |
| General reward sensitivity[+] | Behavioral Approach System Reward Responsiveness (BAS-RR) | Humans Making Music 2 (2023) | 1954 (629) | 1383 (379) | 1510 (363) | 1192 (244) | 2680 (556) | 8719 (2171) |
| Music reward sensitivity | Barcelona Music Reward Questionnaire (BMRQ) | Humans Making Music 2 (2023) | 2025 (659) | 1459 (400) | 1595 (386) | 1258 (268) | 2832 (592) | 9169 (2305) |

We collected the two measures for general perceptual-affective phenotypes (top two rows) and one measure for music reward sensitivity (bottom row) from two data collection waves in a sample of twins from the Swedish Twin Registry. The measures used to quantify each trait are shown in the second column. The third column provides the year of completion of the data collection from the Swedish Twin Registry for the Humans Making Music studies. The number of pairs with data available for both twins (*n* pairs) is shown in parenthesis. Note that the *n* in wave 1 is smaller as we only included twins that also participated in wave 2. *n* number of individual twins, MZ Monozygotic, DZ Dizygotic, os opposite-sex.[+] The total sample size for these traits is shown only for twins in which music reward sensitivity data were available. See Supplementary Table 1 for descriptives. Additionally, note that we found no evidence of sampling bias effects on the BMRQ, yet we found some evidence for the SMDT (see Supplementary Table 2 for a test of participation bias).

fitted the data, as indicated by comparison against the saturated model ($\chi^2(\Delta df = 33) = 41.13$, $p = 0.16$). However, a more parsimonious AE model, from which the D component was dropped, did not significantly worsen the model fit ($\chi^2(\Delta df = 1) = 1.63$, $p = 0.20$). Therefore, the AE model was deemed the best model for the data. The heritability ($h^2_{twin}$) of the BMRQ total score was substantial: $h_{twin}^2 = 0.54$ (95% CI [0.51, 0.58]; Fig. 1d; see Supplementary Table 4 for details).

### Music reward sensitivity is influenced by genetic factors above and beyond genetic influences shared with music perceptual abilities and general reward sensitivity

To better understand the nature of genetic effects contributing to music reward sensitivity, we tested whether the genetic influences on BMRQ were shared with other related traits, such as music perceptual abilities and general reward sensitivity. For this purpose, we used a Cholesky decomposition approach, which allowed us to discriminate between three possible outcomes. Genetic effects on music reward sensitivity could be either fully or partly separate from genetic effects on music perceptual abilities or general reward sensitivity, implying no or only partial genetic overlap across traits. Alternatively, genetic effects could be fully shared and, hence, completely overlapping.

First, we revealed and confirmed that there were significant phenotypic correlations between music reward sensitivity and music perceptual abilities and general reward sensitivity[10,11], $r = 0.19$ (95% CI [0.15, 0.23]) and $r = 0.29$ (95% CI [0.27, 0.32]), respectively ($p < 0.001$). Music perceptual abilities and general reward sensitivity were not significantly correlated, $r = -0.02$ (95% CI [−0.06, 0.03]), $p = 0.44$. (Correlations were estimated from a sample selecting only one twin per pair to avoid sample dependence; estimates were similar in the other twins, see Supplementary Fig. 4 for details.) To simultaneously accommodate the three phenotypes, we employed a tri-variate Cholesky decomposition (Fig. 2a). To test whether the three phenotypes displayed partial overlap between genetic factors, we compared the full Cholesky decomposition with models in which genetic cross-paths from music perceptual abilities or general reward sensitivity to music reward sensitivity were set to zero, respectively. Removing the genetic cross-paths worsened the model fit ($\chi 2(\Delta df = 1) = 94.37$ and $\chi 2(\Delta df = 1) = 87.71$, both $p < 0.001$), indicating partial overlap between genetic factors (see Supplementary Table 5). The $h_{twin}^2$ of music reward sensitivity adjusted for music perceptual abilities and general reward sensitivity was adj-$h_{twin}^2 = 0.38$ (95% CI [0.33, 0.43], Fig. 2b). Thus, of the total variance in music reward sensitivity explained by genetic factors ($h_{twin}^2 = 0.54$), around 70% (95% CI $\sigma_{Au:At}^2 = [0.63, 0.78]$) was unique to this trait. Only the remaining 30% was shared with genetic effects on music perceptual abilities and general reward sensitivity, explaining 12% and 18% of the total genetic variance in music reward sensitivity, respectively. Environmental analyses revealed that only removing the environmental cross-path from general reward sensitivity to music reward sensitivity worsened the fit of the Cholesky ($\chi 2(\Delta df = 1) = 64.07$, $p < 0.001$) while removing the one from music perceptual abilities did not ($\chi 2(\Delta df = 1) = 0.004$, $p = 0.95$). This indicates a small overlap in environmental influences between general reward sensitivity—but not music perceptual ability—and music reward sensitivity, which explained 2% of the total variance in music reward sensitivity (see Supplementary Note 2).

### Genetic pathways to the different facets of music reward sensitivity are partly distinct

Having shown that music reward sensitivity has substantial heritability and is partly genetically separate from relevant general perceptual-affective processes, we went on to test whether the pattern of twin correlations across facets is consistent with largely distinct genetic and environmental influences on the different facets of music enjoyment (Fig. 3a) or with an overarching genetic or environmental shared factor capturing genetic and environmental influences shared across facets

**a**   Saturated model with age as covariate

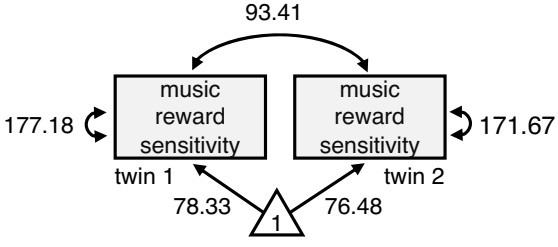

**b**   Twin correlations

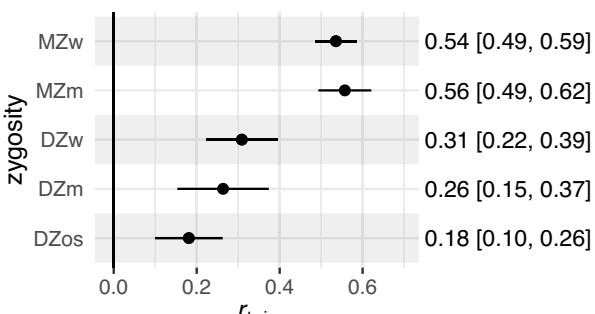

**c**   ADE model

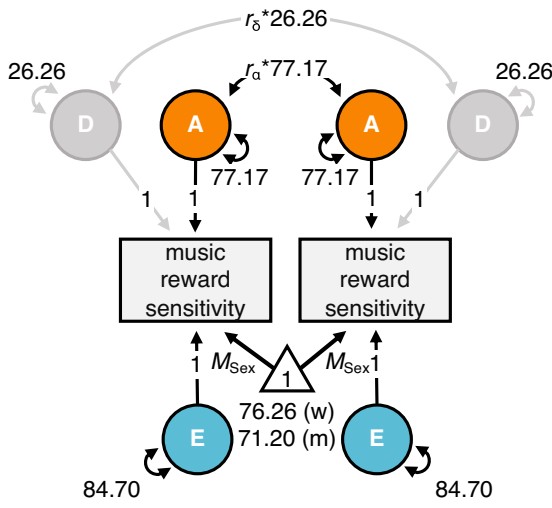

**d**   Implied decomposition of variance

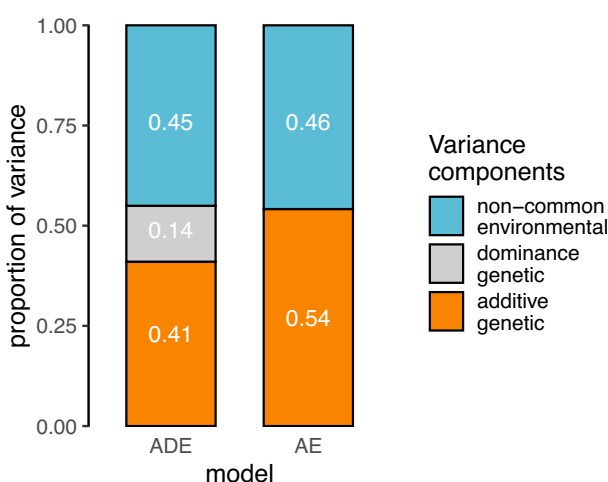

**Fig. 1 | Music reward sensitivity is substantially heritable. a** Saturated model (including age as a covariate; not shown) used for assumption testing and comparisons of CTD-informed models fit; for simplicity, only one group (MZ women) is shown. **b** Twin pair correlations ($r_{twin}$) grouped by zygosity and sex (w women, m men, os opposite-sex, see Table 1, Music reward sensitivity ($n$ pairs), for the sample size of complete twin pairs grouped by zygosity); note that MZ twin pairs are more than twice as similar in their music reward sensitivity as DZ twin pairs. Data are presented as correlation estimates with 95% confidence intervals (CI). **c** The ADE model; note that we identified only additive genetic (A) and non-common environmental (E) components as significant contributors to music reward sensitivity variability (see Supplementary Fig. 3 for the complete diagram). $r_\alpha$ is the expected additive genetic relationship, and $r_\delta$ is the expected dominance genetic relationship between pairs (i.e., $r_\alpha = 1$ or 0.5, $r_\delta = 1$ or 0.25, for MZ and DZ, respectively).

**d** Estimated variance components from the final AE model indicated substantial heritability for music reward sensitivity. The left bar plot shows the estimates obtained from the full ADE model. Model estimates are provided as Source Data. Notes on structural equation models: For simplicity, age is not included in the graphical representation of the model but is included as a covariate; Squares or rectangles represent the measured phenotypes; Circles the latent variable; Double-headed arrows connecting circles with themselves, the variances associated with the latent components; double-headed arrows connected between circles covariances; the triangle indicates the unit vector regressed on the phenotype, which corresponds to the sex-specific phenotypic means, $M_{sex}$ (already adjusted for age); greyed elements, the component dropped after model comparison. A additive genetic, D dominance genetic, E non-common environmental.

(Fig. 3b, c). The first scenario can be modelled as a multivariate correlated factor solution, which solely allows for genetic and environmental pairwise correlations (Fig. 3a). If, on the other hand, there is a shared genetic source of different aspects of musical enjoyment, we would expect underlying genetic sources of variability to be mostly shared across different facets (Fig. 3b, see ref. 43). This latter scenario can be instead modelled as a multivariate hybrid-independent pathway model (see ref. 44). Here, along with distinct genetic effects over single facets, an extra additive genetic shared factor is modelled to capture shared genetic effects across all facets. A similar modelling solution can be applied to test the structure of environmental influences (Fig. 3c). For ease of interpretation, we will hereafter refer to the models depicted in Fig. 3a–c as the distinct factor solution, the shared-genetic factor solution and the shared-environmental factor solution, respectively.

Since the shared-genetic factor solution is a constrained version of the distinct factor solution, model comparison can be used to test whether a shared genetic factor of music reward sensitivity facets shows a better fit to the data. Phenotypic and twin correlations across BMRQ facets data are reported in Table 2. While all three models fit the twin data well (CFI = 0.988, SRMR = 0.048; CFI = 0.985, SRMR = 0.048; and CFI = 0.987, SRMR = 0.048 respectively), both the shared-genetic and shared environmental factor showed worse fit than the distinct factor solution ($\chi^2(\Delta df = 5) = 65.14$, $p < 0.001$; $\chi^2(\Delta df = 5) = 34.125$, $p < 0.001$). Note that model comparisons indicated that dropping the D component from the facets did not significantly worsen the model fit (see Supplementary Tables 6 and 7). In summary, the distinct factor solution is more appropriate than the shared factor solutions (Fig. 4a–c; See Supplementary Note 3 for more details).

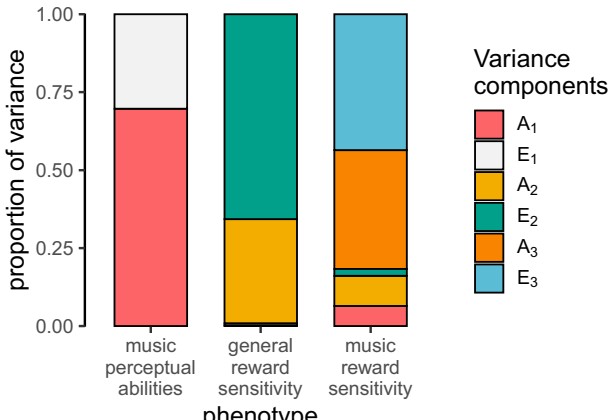

**a** Partial overlap of genetic effects

**b** Implied sequential decomposition of variance

**Fig. 2 | Genetic effects on music reward sensitivity are partly separate from music perceptual abilities and general reward sensitivity. a** Cholesky decomposition of the significant contributions to music reward sensitivity; note that the environmental path from music perceptual abilities to music reward sensitivity is not significant, indicating only overlapping genetic causes (see Supplementary Fig. 5 for the complete path-diagram). Estimates under the square root represent the phenotypic variance explained by the respective variance components. Model estimates are provided as Source Data. **b** Variance decomposition shows that genetic factors (A) explain individual differences in music reward sensitivity (in orange) well beyond shared genetic factors associated with general perceptual and affective processes (in red and yellow, respectively). Environmental factors (E) that contribute significantly to music reward sensitivity in green and turquoise. Notes on structural equation models: one-headed arrow represents regression paths partitioned in additive genetic and non-common environmental paths; greyed one-headed arrows represent non-significant paths. Other abbreviations and symbols are as in Fig. 1.

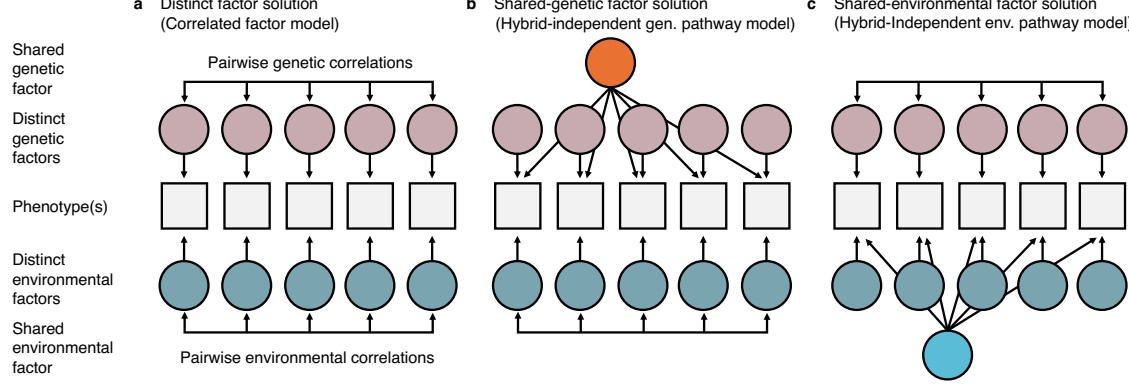

**a** Distinct factor solution (Correlated factor model)

**b** Shared-genetic factor solution (Hybrid-independent gen. pathway model)

**c** Shared-environmental factor solution (Hybrid-Independent env. pathway model)

**Fig. 3 | Schematic illustration of multivariate models employed to quantify distinct and shared genetic and environmental influences on the facets of music reward sensitivity. a** The first solution includes distinct genetic and environmental factors with a simple description of all possible genetic and environmental covariances. **b** A shared-genetic factor solution is applied by assuming a shared latent genetic factor (in orange) that captures all the genetic covariances across facets. This shared genetic factor also partly or fully explains the genetic variance associated with the facets. **c** A similar solution as the one depicted in (**b**), but including a shared factor for the environmental covariances. Note these schematic representations avoid cluttering by compressing all ten possible pairwise correlations in the double-headed arrows; see Supplementary Figs. 6–8 for the full path-diagrammatic representation of the models.

**Table 2 | Phenotypic and twin correlations for the BMRQ facets**

| BMRQ facets | Phenotypic correlations | | | | | Twin correlations (MZ\|DZ) | | | | |
|---|---|---|---|---|---|---|---|---|---|---|
| | f1 | f2 | f3 | f4 | f5 | f1 | f2 | f3 | f4 | f5 |
| Emotion evocation (f1) | 3.47 | - | - | - | - | 0.44\|0.18 | 0.12 | 0.12 | 0.11 | 0.17 |
| Mood regulation (f2) | 0.61 | 3.29 | - | - | - | 0.32 | 0.43\|0.17 | 0.14 | 0.11 | 0.15 |
| Music seeking (f3) | 0.44 | 0.58 | 3.85 | - | - | 0.24 | 0.34 | 0.48\|0.18 | 0.09 | 0.15 |
| Sensory motor (f4) | 0.42 | 0.42 | 0.39 | 3.61 | - | 0.23 | 0.23 | 0.23 | 0.53\|0.20 | 0.12 |
| Social reward (f5) | 0.53 | 0.54 | 0.48 | 0.48 | 3.58 | 0.31 | 0.32 | 0.28 | 0.31 | 0.52\|0.26 |

Left-hand columns display BMRQ facets (f) phenotypic correlations. Standard deviations of facets are shown on the diagonal. Right-hand columns display both within- and between-trait twin correlations, with MZ\|DZ within-trait correlations on the diagonal. MZ between-trait twin correlations (below diagonal) are about twice as high as between-trait DZ twin correlations (above diagonal) across all facets. See Source Data for observed correlations stratified across all five sex-specific zygosity groups.

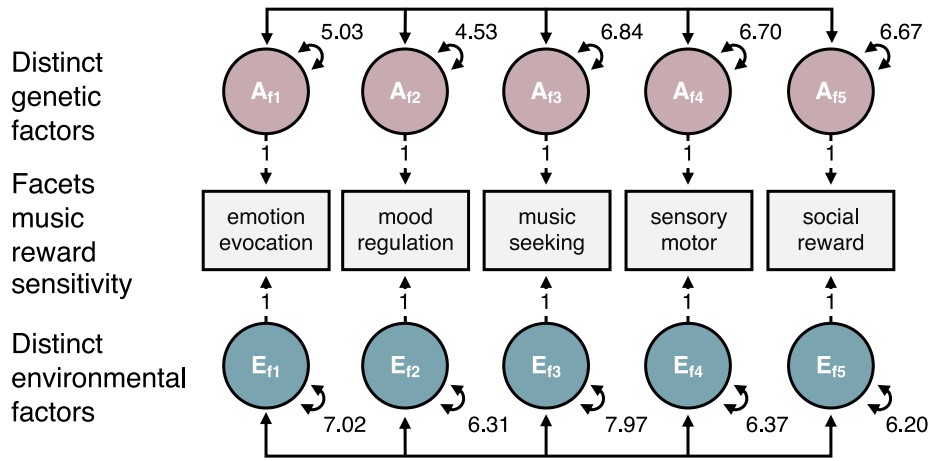

**a**  Multivariate model comparison favours distinct factor solution

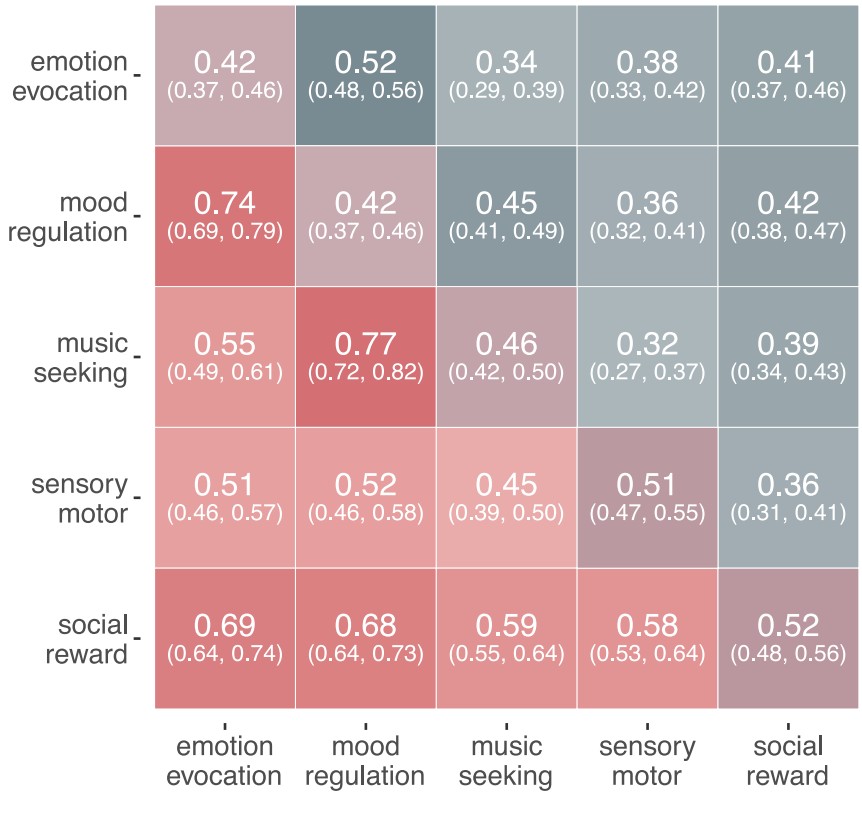

**b**  Genetic and environmental correlations

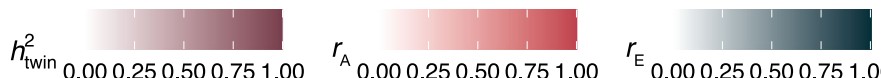

**Fig. 4 | Genetic effects on music reward sensitivity are partly heterogeneous. a** The best-fitting correlated factor model (see Supplementary Fig. 6 for the complete path-diagrammatic representation and Source Data for model estimates). **b** Matrix extracted from the best-fitting correlated factor model. Additive genetic ($r_A$) and environmental correlations ($r_E$) are shown below (reddish) and above (greyish) the diagonal, respectively; numbers on the diagonal show heritability ($h_{twin}^2$) estimates. Numbers in parentheses are 95% CI. Note that genetic correlations are high but far from 1, suggesting that music reward sensitivity has multiple partially shared genetic sources. The percentage of phenotypic covariance explained by genetic and environmental effects can be found in Supplementary Fig. 9. Notes on structural equation models: double-headed arrows between circles represent all possible pairwise A and E covariance between facets. Other abbreviations and symbols are as in Figs. 1 and 3.

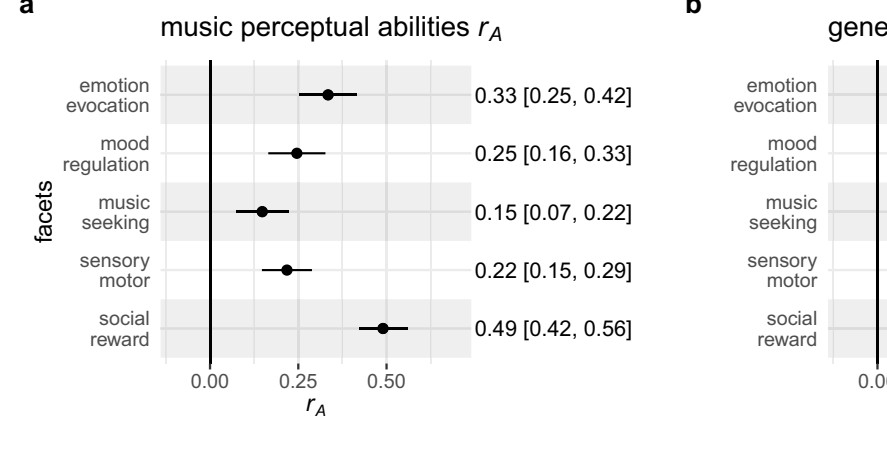

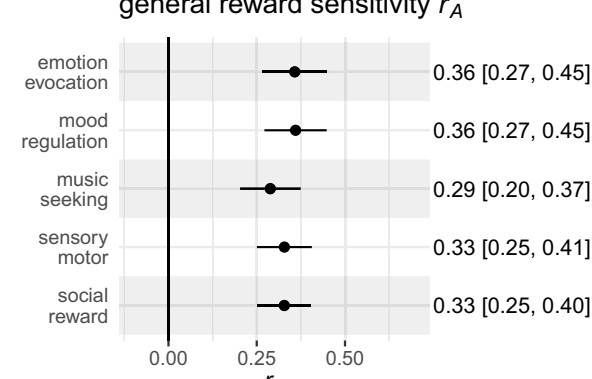

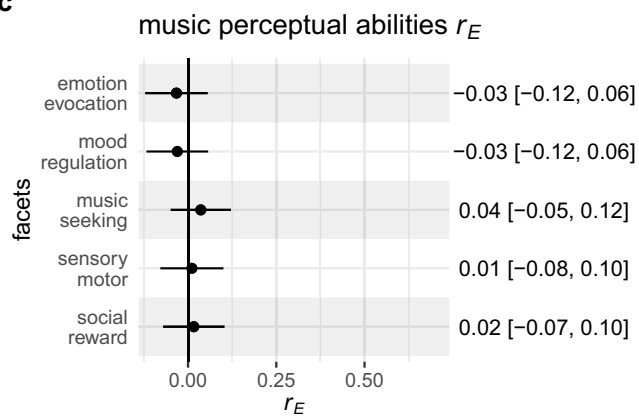

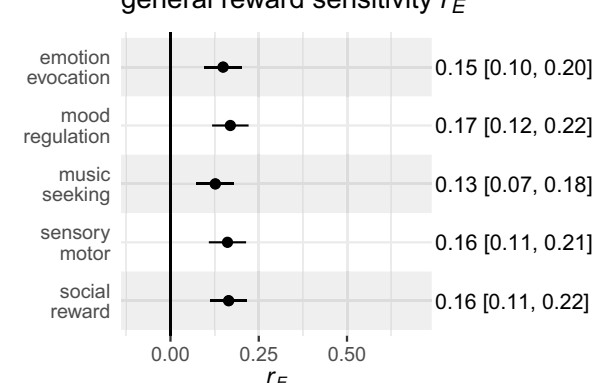

**Fig. 5 | Disentangled genetic and environmental correlations between music perceptual abilities, general reward sensitivity, and music reward sensitivity facets.** Diagrams show genetic ($r_A$) and environmental ($r_E$) correlations from the two extended six-variate distinct factor solutions. **a, b** Magnitude of $r_A$ between facets of music reward sensitivity, music perceptual abilities, and general reward sensitivity. Note that in (**a**), the $r_A$ between music perceptual abilities and the social-reward facets are stronger than for every other facet. **c, d** Magnitude of the environmental correlations ($r_E$). Data are presented as correlation estimates with 95% CI. See Table 1, Music perceptual abilities and General reward sensitivity (*n* pairs), for the sample size of complete twin pairs grouped by zygosity and Supplementary Table 8 for details.

**Musical perceptual abilities share more genetic variance with social reward than with any other facet of musical reward**

We further explored the heterogeneity in musical reward by fitting two additional six-variate distinct factor solutions including music perceptual abilities, general reward sensitivity, and music reward sensitivity facets. Additive genetic correlations ($r_A$) between music perceptual abilities and music reward sensitivity facets varied widely (range $r_A = 0.15$ to $r_A = 0.49$; Fig. 5a). The differences between the genetic covariances underlying these $r_A$ values were significant. Specifically, the $r_A$ estimates were higher for the social-reward facet of music reward ($r_A = 0.49$, 95% CI [0.42; 56]) than for any other facet (range $\Delta r_A$ from 0.19 to 0.39, all *p*-values < 0.001). In contrast, genetic correlations with general reward sensitivity were of similar magnitude (range $r_A = 0.29$ to $r_A = 0.36$; Fig. 5b), with no significant differences between facets (all *p*-values for the differences > 0.10). These observations further strengthen the conclusion that different aspects of music reward show functionally relevant genetic heterogeneity. Finally, environmental correlations ($r_E$) between music perceptual abilities and music reward facets were all non-significant (*p*-values > 0.40, Fig. 5c), while $r_E$ values between general reward sensitivity and music reward sensitivity facets were all significant (*p*-values < 0.001). However, they were all of similar magnitude (range $r_A = 0.13$ to $r_A = 0.17$; Fig. 5d) and did not significantly differ from each other (all *p*-values for the differences > 0.11).

## Discussion

Our understanding of why "meaningless tonal patterns"[8] have such powerful effects on humans can benefit tremendously from the study of inter-individual differences. Here, by exploiting a large Swedish twin sample, we found that music reward sensitivity has substantial twin-based heritability. Most of this genetic variance influences music reward sensitivity independently of music perceptual abilities and general reward sensitivity, suggesting that genetic variants influence music reward sensitivity not only via other general perceptual-affective processes. Furthermore, our findings reveal considerable genetic heterogeneity behind different facets of music reward sensitivity. Although all facets show heritability estimates of a similar magnitude (between 42% and 52%) and are genetically correlated (between 0.45 and 0.77), the results do not support a single (genetic or environmental) dimension of musical enjoyment. Instead, these findings are consistent with individual differences in musical enjoyment being built upon genetically interconnected, yet partly distinct components. Extended multivariate analyses further strengthened this conclusion by showing that music perception has stronger genetic correlations with social bonding than with other facets of music reward, indicating functionally relevant genetic heterogeneity.

Despite answering the long-standing question of whether individual differences in music enjoyment are heritable[28,29], the finding that music reward sensitivity is to some extent heritable is not surprising in

light of the fact that virtually every human trait is at least partly genetically influenced[45,46]. Yet, the finding of a 54% heritability for music reward sensitivity gives hope for molecular genetic studies to answer questions about genetic underpinnings of musicality in general, and musical affect in particular. Prior studies of individual differences in music reward sensitivity[11,14,15,47,48] have had far-reaching implications for our knowledge of biological pathways implicated in perceptual-affective processes[17–19]. These studies have shown that individual differences in music reward sensitivity are associated with variations in functional and structural connectivity between two systems. The first includes the auditory cortex and associated pathways involved in perceptual analysis, feature encoding, working memory, and prediction. The second, the reward system, encompasses the striatum, orbitofrontal cortex, and ventral tegmental area, and is involved in pleasure, salience, and learning[1,15,18,19,49,50]. These neurobiological mechanisms could provide a potential substrate for the genetic influences identified in the present study. Therefore, an important question for future studies is to investigate whether variability in structural and functional properties of the relevant brain networks and their interactions, may mediate the genetic effects on the ability to enjoy music, thus furthering our overall mechanistic understanding of an important aspect of human affect.

A complementary genetic perspective on music reward sensitivity could be a particularly fruitful strategy to better understand human musicality and affect because we found genetic influences to be in part separate from other relevant perceptual-affective processes, such as music perceptual abilities and general reward sensitivity. The dissociation between the genetics of music reward sensitivity and general perceptual and reward processing mirrors the finding that specific musical anhedonia, i.e., blunted or absent hedonic responses from music stimuli, exists in the absence of any perceptual or generalised reward deficit[11,15]. This observation implies that genetic variance associated with music reward sensitivity, beyond perceptual and general reward processing, can be used to disentangle and better understand the mechanisms involved in sensory-specific experiences of enjoyment.

The partial separation between genetic effects on perception and enjoyment is consistent with the possibility that genes influencing music perception and enjoyment may have been separately targeted for natural selection during evolution[51]. The finding implies that genetic variation between people may be used to dissect the evolutionary trajectories of different aspects of human musicality. Along these lines, a further question of interest becomes whether genetic variants, which are more specifically associated with music enjoyment, are also enriched in genomic regions of evolutionary interest[52,53].

Here, we did not find support for a single overarching shared genetic factor of music reward sensitivity, suggesting no single common intermediate pathway from genes to different aspects of music enjoyment (with pathway, we refer to the combination of processes along the complex chain from genetic to phenotypic differences as in ref. 54). On the contrary, we found partly distinct and partly overlapping genetic pathways to music enjoyment. This result aligns with general views of musicality as "built upon a suite of interconnected capacities, of which none is primary"[55]. Our results demonstrate that such heterogeneity is seen even when zeroing in on one hypothesised core feature of musicality: enjoyment. We show that music reward sensitivity is itself not a monolith and that different facets of this trait are influenced by partly different genetic pathways; these facets range from the ability to experience emotion and get chills to the rewarding aspects of social bonding through music. Our results may indicate that future investigations on the genetic contribution to individual differences in music enjoyment should focus on the separate facets rather than the total music reward sensitivity score.

Our final analysis provides a direct example of the implications such a shift in perspective might have for the study of human

behaviour and affect. When dissecting the genetic effects at the level of the facets of music reward sensitivity, insights emerge. Our findings indicate that music perceptual abilities are genetically more strongly correlated with rewards of social bonding through music, as compared to other facets of music reward sensitivity. This could be seen as in line with the social bonding hypothesis, which states that "core biological components of human musicality evolved as mechanisms supporting social bonding"[56]. This was not the case for the genetic and environmental association between music reward and general reward sensitivity, which were relatively similar across different facets. Furthermore, shared additive genetic variation entirely explained the association between music perceptual abilities and social reward, suggesting shared biological components to be at play. These results highlight how acknowledging the genetic heterogeneity of music reward sensitivity might reveal associations that might have been otherwise unnoticed. (For a detailed discussion on consequences for other well-studied conditions, such as musical anhedonia[11,14,15], we refer to Supplementary Note 4.)

Notwithstanding such functionally relevant genetic heterogeneity, we also found genetic overlap between the facets, suggesting that genetic effects over music reward sensitivity are also partially shared. This finding is important as some degree of genetic overlap across facets of music reward sensitivity is needed to better understand the biology of music enjoyment as a whole. Further studies could test whether these genetic effects underlie other auditory phenomena, such as pleasure derived by timbre in sounds, which has been shown to correlate homogenously across facets of music reward[30], or other broader aspects related to human affect, such as aesthetic sensitivity[57].

We also show that environmental influences broadly mirror genetic influences on music reward sensitivity and its facets. Although effects explained a similar magnitude of the variance in music reward sensitivity facets (48%–58%), we also found no support for a single shared environmental factor common across facets of musical enjoyment. Yet, compared with genetic overlap, we found notable exceptions: First, environmental effects on music perceptual abilities and music reward sensitivity were non-overlapping. Second, environmental correlations between general and music reward sensitivity were similar. Although it would be tempting to speculate on why this is the case, it is challenging to interpret environmental effects and correlations as the environmental component includes all residual sources of variability left after accounting for genetic effects, including measurement error. Furthermore, music perceptual abilities and general reward sensitivity were collected at different time points, possibly attenuating correlations between the two traits and making it more difficult to draw inferences on differences in non-common environmental effects.

Finally, the absence of shared environmental effects on music reward sensitivity in an adult sample (at least under the assumption of the classical twin design, see below) aligns with many other complex traits, including those related to musicality[27,45]. Yet, it contrasts with findings on other musicality traits, such as musical achievement[22,25] or singing abilities[58], for which modest effects of shared environment have been found using similar designs. The lack of shared environmental effects in adulthood for some traits but not others suggests that different aspects of musicality, namely producing music and enjoying music, might follow different patterns of cultural and genetic intergenerational transmission. The likely absence of shared environmental effects may imply that passive gene-environment correlation (i.e., indirect genetic effects, which are environmentally mediated[59]) plays little role in music reward sensitivity in adulthood. This is crucial because passive gene-environment correlations would complicate future efforts to detect direct genetic effects on music reward sensitivity by, e.g., confounding direct genetic effects with indirect effects (see refs. 59,60 for a detailed discussion). Recent efforts to better understand the genetic architecture of complex traits focus on

deconstructing indirect sources of heritability, which inflate estimates of genetic effects and confound the possible inferences that can be obtained from downstream analysis of genome-wide-derived estimates[60–62]. Our findings suggest that music reward sensitivity, or rather its constituent facets, may be especially promising for facilitating discoveries of direct molecular genetic effects on music enjoyment in adults.

As with every other twin-informed study[38], our work depends on a number of assumptions. In the "Methods" section, we highlight these assumptions and what violation of each entails. One critical assumption is the equal environment assumption, which states that environmentally caused differences between twins within a pair are the same across zygosities. However, we also note that the equal environment assumption is not violated if different zygosities experience more similar or dissimilar environments due to genetic differences. On the contrary, this is to be expected if evocative and active gene-environment correlations are at play, which seems likely for traits related to music enjoyment. Such gene-environment correlations may change the interpretation of the heritability estimates as they could reflect, for example, a more complex causal chain that leads individuals to seek or be exposed to environmental changes that, in turn, influence the phenotype, amplifying genetic effects via processes such as niche picking[63,64]. An additional assumption is the lack of gene-by-shared environment interaction, which could lead to an overestimation of the variance of the A component. The lack of common environmental effects in adulthood, as found in this study, does not imply a lack of common environmental effects on similar musicality traits earlier in the lifespan. On the contrary, common environmental effects seem to contribute to variability in musicality traits in younger children[24], implying possible passive gene-by-common environment correlation (e.g., parents may both pass genetic variants and provide musically enriched environment contributing to music reward sensitivity of their children) or interactions (e.g., additive genetic effects associated with music reward sensitivity might vary within different musically enriched environments that the family provide to the children) at earlier stages in life. As such, an important avenue for future research will be to quantify whether gene-environment correlations or interactions are at play in music reward sensitivity at earlier stages in life and how much they impact quantitative genetic estimates for musicality traits more broadly. Further, although we did not detect significant non-additive (i.e., dominance) genetic effects on music reward sensitivity and its facets, our estimates for twin-based heritability may represent broad-sense heritability as any non-additive genetic variance would be pushed into the estimate for the variance of the A component.

Future research should consider two additional important aspects. First, although we found no evidence of sampling bias on music reward sensitivity, we also found evidence that individuals with lower-than-average music perceptual abilities tend to be slightly underrepresented in our study (see Supplementary Table 2). This finding is not completely surprising, as population-based studies based on voluntary participation, such as ours, are known to be subject to participation bias (e.g., ref. 65). Nevertheless, the effect was small and evident only for a secondary measure in this study. Second, although a recent study in Norwegian twins reported a similar heritability estimate for sensitivity to music[66], we caution against a generalisation of our results to non-Northen European populations[67]. It is important to keep in mind that heritability estimates may be both population- and environment-specific (for misconceptions about heritability, see ref. 68). Here we have focused our analysis on a particular musicality trait shown to be present in diverse populations[10,69–71]. Whether the genetics of individual differences in music enjoyment are similar across populations is yet to be studied.

Much has been said about the sources of the considerable inter-individual variation in music perception, production, participation,

and achievement. Yet, relatively little has been written on the genetic contribution to what makes individuals differ in their capacity to enjoy music. Here, we added a piece to the puzzle of why music can have such powerful effects on humans. We show that the heritability of the ability to enjoy music is largely distinct from the heritability of other, more general aspects of perceptual and affective processing. Further, we reveal that genetic and environmental contributions to individual differences in music enjoyment are partially distinct and that the genetic overlap between music perceptual abilities differs between different facets of music reward. In summary, the findings highlight the complex and multifaceted nature of music enjoyment and its genetic underpinnings, paving the way for further studies of the evolutionary origins and neurogenetic mechanisms for an important aspect of human affect.

## Methods

### Ethics
Both waves of data collection (see below) were approved by the Regional Ethical Review Board in Stockholm (Dnrs 2011/570-31/5, 2012/1107/32, 2021-02014, 2022-00109-02, 2020-02575). All methods and procedures followed international guidelines in accordance with the Declaration of Helsinki.

### Sample
**Swedish Twin Registry: Screening Twin Adults Genes and Environment (STAGE).** Participants were twins recruited from the Swedish Twin Registry[72]. Twin zygosity was determined by questionnaire data, which, when compared to genotypes, has been shown to be 99% accurate in the Swedish Twin Registry[73]. The twins included in this study took part in two large recent waves of online data collection on music, art, and cultural engagement. In 2011 and then again in 2022, a total of 32,005 adult twin individuals were invited from the STAGE cohort born between 1959 and 1985, of which around 11,500 participated in the first wave and then around 9500 in the latest wave. In the first wave of data collection, participants completed the Swedish Musical Discrimination Test (see below). More details on this wave can be found in Ullén et al.[12]. Additionally, in the second wave, the Behavioral Approach System[39] and Barcelona Music Reward Questionnaire[10] were administered. We note that response rates in the STAGE cohort have been low (-30%). The low response rates presumably reflect that this is a population-based study (i.e., where a whole birth cohort of Swedish twins is invited to participate) of a working-aged cohort; this is a general phenomenon not unique to our study[72]. A full description of the twin sample across waves of data collection can be found in Table 1, including *n* of twins for which we had both data available, stratified by the zygosity and the sex of the twins; for both waves of data collection, informed consent was given by each participant before data gathering began.

### Primary measure
**Barcelona Music Reward Questionnaire (BMRQ).** The Barcelona Music Reward Questionnaire (BMRQ) is a psychometric tool used to assess musical anhedonia[11,15] and, more generally, music reward sensitivity[10], which has previously been validated across many cultures[10,69–71]. It comprises 20 self-report items, with five response options, ranging from completely disagree to completely agree. After recoding response items to numeric options (1–5), with two out of 20 items being reverse coded, we used the sum score of the BMRQ as a measure of music reward sensitivity (score range from 20 to 100). Following the original five-factor structure[10], we also created sum scores of the five known facets of music reward sensitivity[29]: (1) Emotion evocation - the degree to which individuals get emotional, experience chills, and even cry when listening to music; (2) Mood regulation - the degree to which individuals experience rewards from relaxing when listening to music; (3) Musical seeking – the pleasure

associated with the discovery of novel music-related information; (4) Sensory motor – the rewards obtained from synchronising to an external beat or dancing; (5) Social reward – the rewards of social bonding through music. Additional details are given in Supplementary Note 5.

### Secondary measures

**Behavioral Approach System Reward Responsiveness (BAS-RR).** The Behavioral Approach System (BAS) scale is included in the Behavioral Inhibition System (BIS)/BAS questionnaire, a validated psychometric tool to assess inter-individual differences in two general motivational systems[39,74]. The BAS-Reward Responsiveness (BAS-RR) scale, in particular, assesses inter-individual differences in the ability to experience pleasure in the anticipation and presence of reward-related stimuli and predicts general psychological adaptive functioning[75]. It comprises five items, with four response options for each. BAS-RR is obtained by the sum score of the five items after the numerical conversion of the responses (1–4). Additional details are given in Supplementary Note 6.

**Swedish Musical Discrimination Test (SMDT).** The Swedish Musical Discrimination Test (SMDT) is a test that has good psychometric qualities for individual abilities in auditory perceptual discrimination of musical stimuli[12]. It comprises three subtests: melody, rhythm, and pitch. A brief description of each test is given below (see ref. [12] for more details).

**Melody.** This subtest used isochronous sequences of piano tones as stimuli. Tones ranged from C4 to A#5, played at 650 ms intervals (American standard pitch; 262–932 Hz). The number of tones increased from four to nine during the subtest progression. For each of the six stimulus lengths, there were three items. The two stimuli in an item were separated by 1.3 s of silence. The pitch of one tone in the melody was always different in the second stimulus. Participants had to identify which tone was different.

**Rhythm.** In this subtest, each item included two brief rhythmic sequences of 5–7 sine tones, lasting 60 ms each. The inter-onset intervals between tones in a sequence were 150, 300, 450, or 600 ms. The two sequences in an item were either identical or different and separated by 1 s of silence. The participant had to determine whether the two sequences were the same or not.

**Pitch.** The pitch subtest used sine tones with a 590 ms duration as stimuli. In each item, two tones were presented, one of which always had a frequency of 500 Hz. The frequency of the other tone was set between 501 and 517 Hz. The order of the two tones varied randomly, with tones separated by a 1 s silence gap. Participants had to identify whether the first or the second tone had the highest pitch. The item difficulty was increased progressively by gradually making the pitch differences between the tones smaller.

### Analyses

**Threshold for statistical significance (α value).** Throughout the whole study, we used an adjusted alpha of $\alpha = 0.007$. The adjusted alpha was obtained via the Bonferroni correction as $\alpha = 0.05/M_{eff}$. $M_{eff} = 7$ is the number of effective tests accounting for dependency between variables and was obtained, following[76], as the number of eigenvalues required to explain 99.5% of the variance across all the variables included in this study. (The correlation matrix included seven variables, i.e., the SMDT and BAS-RR total scores and the five BMRQ facets' scores; we have excluded the BMRQ total score as it is a linear combination of the five BMRQ facets' scores.) $M_{eff}$ was computed using the meff (method = "gao") function from the poolr R package[77]. Alternative methods (e.g., Li & Ji[78]) resulted in a less conservative alpha (i.e., $M_{eff} < 7$).

**Factor analysis.** To confirm the BMRQ's sum score as an appropriate measure of music reward sensitivity in the Swedish sample, we ran a one-factor Confirmatory Factor Analysis (CFA) on the five facets of the Swedish version of the BMRQ. CFA was run on one twin per pair, using the lavaan::cfa() function, to avoid sample dependence.

**Classical twin design (CTD).** The CTD allows the estimation of additive (A) or dominance (D) genetic, common environmental (C), and residual source (E) of phenotypic variance ($\sigma_A^2$, $\sigma_D^2$, $\sigma_C^2$, and $\sigma_E^2$, respectively). This is possible given the expected phenotypic resemblance of monozygotic (MZ) and dizygotic (DZ) twins. MZ twins arise from the same fertilised egg and thus are ~100% genetically similar (with minimal deviations from expected genetic similarity, see ref. [79]); DZ twins arise from separate egg cells and thus, as ordinary siblings, share on average 50% of their segregating genes. Furthermore, when both twins of a pair are raised in the same household, MZ and DZ twins share 100% of their common environment. Finally, by definition, remaining deviations from the expected values inferred by additive, dominance, and common environmental effects represent non-common environmental influences and measurement errors. Therefore, E is not shared between twins within a family. Under a set of assumptions, including no epistasis (gene-by-gene interaction, see ref. [80]), the covariance of MZ twin pairs is then equal to:

$$\sigma_{MZ,MZ} = \sigma_A^2 + \sigma_D^2 + \sigma_C^2 \qquad (1)$$

While the covariance of DZ twin pairs is equal to:

$$\sigma_{DZ,DZ} = 0.5 * \sigma_A^2 + 0.25 * \sigma_D^2 + \sigma_C^2 \qquad (2)$$

Given that the variance and covariance are measured between twins within families, it is possible to specify a multigroup structural equation model and estimate three out of four variance components. The decision of which parameters to include in the model (e.g., A, C, E, or A, D, E) is purely based on twin covariances, which are extracted from the constrained saturated model phenotypic model (see below), and biological plausibility. If $\sigma_{MZ,\ MZ} > 2*\sigma_{DZ,DZ}$, then D is expected to contribute to the phenotypic variance and, therefore, an ADE model is specified. Otherwise, an ACE model is fit to the data.

**CTD assumptions.** The estimates from the CTD are unbiased under a set of assumptions. First, the CTD assumes equal environments between the twins. In other words, it assumes that similarities between twins caused by the environment are the same for both zygosities. Suppose, instead, MZ twins experience their environment more similarly than DZ twins due to environmental, not genetic, causes. In that case, the estimate for the genetic variance will be upwardly biased (i.e., $\widehat{\sigma_A^2} > \sigma_A^2$). Note that the equal environment assumption is not violated if MZ twins experience their environment more similarly than DZ twins due to genetic differences. The second assumption is that the phenotypes of the parents of the twins are uncorrelated (i.e., random mating, also known as panmixia[81]). If the covariance between two parental phenotypes, $p_1$ and $p_2$, is different from 0, $\sigma_{P1,P2} \neq 0$, then the shared environmental variance might be upwardly biased (i.e., $\widehat{\sigma_C^2} > \sigma_C^2$). The third assumption is that there are no gene-environment interactions or gene-environment passive correlations. Based on the type of gene-environment interaction or correlation, different sources of bias are expected. If AxC is present, then $\widehat{\sigma_A^2} > \sigma_A^2$ is expected. If AxE is present instead $\widehat{\sigma_E^2} > \sigma_E^2$. If passive $r_{G,E}$ is present, then $\widehat{\sigma_C^2} > \sigma_C^2$ is expected. An additional set of assumptions introduced when estimating parameters via SEM is that means and variances are equal across zygosity group, twin order (i.e., 1 and 2), and sex. Details on the latter set of assumptions are given below. Complex sources of upward or downward biases in CTD-informed models (e.g., heterogeneity) are discussed elsewhere[82].

**Saturated model.** We first fit multigroup SEM models to create a baseline against which to compare the fit of univariate and multivariate models and test for the assumptions of the equality of mean and variances. The models freely estimated all the observed variance and covariances and included the age of the twins as a covariate. For the univariate model, equality of means and variances was tested by sequentially constraining parameters and comparing the Akaike Information Criterion (AIC) and Bayesian Information Criterion (BIC) of the models, where $AIC = 2k - 2\ln(\hat{L})$ and $BIC = k\ln(k) - 2\ln(\hat{L})$, with $k$ being the number of parameters estimated in the model and $\hat{L}$ the maximised value of the likelihood function. Models with smaller AIC and BIC were deemed a good fit. Additional comparisons are provided by the likelihood-ratio test (LRT), using the lavaan::lavTestLRT() function from the lavaan R package[83]. All models were specified following lavaan notation and fitted with the lavaan::sem() function.

**Univariate variance decomposition.** The SEM specification was informed by the CTD, following the pattern of twin pairs correlations extracted from the SEM model selected after model comparison results. Twin pairs correlations were extracted using the most parsimonious constrained saturated model using the lavaan::standardizedSoultion() function. Precisely, we fit a five-group ADE SEM model, where the five groups were formed by either full or incomplete MZ women, MZ men, DZ women, DZ men, and DZ opposite-sex pairs. Means for women and men were estimated freely across sex, but not across zygosities or twin order. We fit the model via the direct symmetric approach by directly estimating the variances, as it can derive asymptotically unbiased parameter estimates and is, therefore, less prone to type I errors[84]. We then decomposed the variance-covariance matrix **T** of twin pairs into the **T** = **A** + **D** + **E** variance covariances, which was predicted as follows:

$$\mathbf{T} = \begin{bmatrix} \sigma_A^2 + \sigma_D^2 + \sigma_E^2 & r_\alpha * \sigma_A^2 + r_\delta * \sigma_D^2 \\ r_\alpha * \sigma_A^2 + r_\delta * \sigma_D^2 & \sigma_A^2 + \sigma_D^2 + \sigma_E^2 \end{bmatrix} \quad (3)$$

Where $r_\alpha$ is the expected additive genetic relationship, and $r_\delta$ is the expected dominant genetic relationship between pairs (i.e., $r_\alpha = 1$ or $0.5$, $r_\delta = 1$ or $0.25$, for MZ and DZ, respectively). Note that for simplicity, here we exclude the contribution of age to **T**, which was instead included in the model. To test for the significance of the variance components A and D, we additionally fit two models where D and AD variances were constrained to 0. Significance was inferred by model comparison, as above. We fit the model to the raw sum score of the BMRQ using the lavaan::sem() function. Assuming data within pairs were missing at random, we used the recommended estimator for twin data analysis, the full-information maximum likelihood (FIML; argument estimator = "ML"). We used the following estimator for the heritability:

$$h_{twin}^2 = \frac{\sigma_A^2}{\sigma_A^2 + \sigma_E^2} \quad (4)$$

Here we note the detail that $\sigma_A^2 + \sigma_E^2 \neq \sigma_P^2$, as $\sigma_P^2 = \sigma_A^2 + \sigma_E^2 + B^2 * \sigma_{Age}^2$. We also note that, since the E component includes residual deviation, $\sigma_E^2 = \text{inter-}\sigma_E^2 + \text{intra-}\sigma_E^2$, where $\text{inter-}\sigma_E^2$ is the inter-individual variance, and $\text{intra-}\sigma_E^2$ is the intra-individual variance[81]. Comparisons with standard OpenMX protocols are given in Supplementary Note 7 (note that the small differences in test statistics did not lead to different conclusions). A graphical representation of the full univariate multigroup model can be found in Supplementary Fig. 3.

**Cholesky decomposition.** We applied a Cholesky decomposition to SMDT, BAS-RR, and BMRQ twin data[85]. Following the CTD, we specified a multivariate model to estimate path (e.g., $\lambda_{A1}$ and $\lambda_{E1}$) and cross-path (e.g., $\lambda_{A12}$ and $\lambda_{E12}$) coefficients. The predicted $6 \times 6$ **S** symmetric matrix included the within-twin variance covariances on the $\mathbf{S}_{1:3,1:3}$ and $\mathbf{S}_{4:6,4:6}$ elements and the between-twin variance covariances on the $\mathbf{S}_{4:6,1:3}$ and $\mathbf{S}_{1:3,4:6}$ elements. The within-twin variance-covariance matrices were obtained as $\mathbf{A_w} = \mathbf{XX}^T$ or $\mathbf{E_w} = \mathbf{ZZ}^T$, where **X** and **Z** are the lower triangular matrices with the path (on the diagonal) and cross-path (on the off-diagonal) coefficients for the additive genetic and environmental components, respectively. The between-twin variance-covariance $\mathbf{A_b}$ matrix was obtained similarly but considered the expected additive genetic correlations, $r_\alpha$, between MZ or DZ twins. The sequence of variables was purely chosen to regress out $A_1$ and $A_2$, and $E_1$ and $E_2$, respectively, implied from SMDT and BAS-RR observed scores from the BMRQ. To estimate an adjusted heritability (here, for simplicity, adj-$h_{twin}^2$), we calculated the proportion of variance of the BMRQ covarying with the component A over the total BMRQ variance (minus the variance in BMRQ covarying with age):

$$adj - h_{twin}^2 = \frac{\lambda_{A3}^2}{(\lambda_{A3}^2 + \lambda_{E3}^2) + (\lambda_{A13}^2 + \lambda_{E13}^2 + \lambda_{A23}^2 + \lambda_{E23}^2)} \quad (5)$$

Where the numerical subscripts simply indicate the order of the phenotype in the model (e.g., 3 is the BMRQ). To calculate the amount of additive genetic variance unique and associated with BMRQ beyond SMDT and BAS-RR ($\sigma_{Au:At}^2$; u = unique, t = total), we computed the proportion of genetic variance over the total BMRQ additive genetic variance as follows:

$$\sigma_{Au:At}^2 = \frac{\lambda_{A3}^2}{(\lambda_{A3}^2) + (\lambda_{A13}^2 + \lambda_{A23}^2)} \quad (6)$$

A graphical representation of the full multivariate model can be found in Supplementary Fig. 5. Similar to what was reported above, we fit the models using the lavaan::sem() function (estimator "ML") but to standardised variables and latent components. We used LRT for the significance of the $\lambda$ coefficient, where the full model was compared to a model in which any tested $\lambda$ coefficient was set to be equal to 0.

**Distinct factor solution.** To estimate the genetic and environmental correlations between facets of music reward, we applied a correlated factor model via a direct symmetric approach[84] (referred to as a distinct factor solution). The direct symmetric approach is conceptually similar to a correlated factor solution. In the correlated factor solution, the multivariate phenotypic variance-covariance matrix **M** is obtained as $\mathbf{M} = \mathbf{A} + \mathbf{E}$ (in the simplest case of an AE model), with $\mathbf{A} = \mathbf{XR_AX}^T$ and $\mathbf{E} = \mathbf{ZR_EZ}^T$, where **X** and **Z** are the diagonal matrix of the standard deviation $\sigma_A$ and $\sigma_E$ and $\mathbf{R_A}$ is the genetic correlation matrix. Within a direct symmetric approach, instead, a different parametrisation is specified to directly estimate the **M** $10 \times 10$ symmetric matrix as $\mathbf{M} = \mathbf{A} + \mathbf{E}$:

$$\mathbf{M} = \begin{bmatrix} \sigma_{A1}^2 + \sigma_{E1}^2 & \sigma_{A1,A2} + \sigma_{E1,E2} & \cdots & r_\alpha * \sigma_{A1,A4} & r_\alpha * \sigma_{A1,A5} \\ \sigma_{A1,A2} + \sigma_{E1,E2} & \sigma_{A2}^2 + \sigma_{E2}^2 & \vdots & r_\alpha * \sigma_{A2,A4} & r_\alpha * \sigma_{A2,A5} \\ \vdots & \vdots & \ddots & \vdots & \vdots \\ r_\alpha * \sigma_{A1,A4} & r_\alpha * \sigma_{A2,A4} & \vdots & \sigma_{A4}^2 + \sigma_{E4}^2 & \sigma_{A4,A5} + \sigma_{E4,E5} \\ r_\alpha * \sigma_{A1,A5} & r_\alpha * \sigma_{A2,A5} & \cdots & \sigma_{A4,A5} + \sigma_{E4,E5} & \sigma_{A5}^2 + \sigma_{E5}^2 \end{bmatrix} \quad (7)$$

Where the $\mathbf{M}_{1:5,1:5}$ and $\mathbf{M}_{5:10,5:10}$ elements include the within-twin variance and between-traits covariances and are constrained to be equal across zygosities, and the $\mathbf{M}_{5:10,1:5}$ and $\mathbf{M}_{1:5,5:10}$ elements include the between-twin additive genetic within- and between-trait covariances and the expected additive genetic relationship $r_\alpha$, which is fixed to either 1 or 0.5 in MZ and DZ groups, respectively. While this approach may return out-of-bound values, the absence of boundaries has been shown to yield asymptotically unbiased parameter estimates and correct type I and type II error rates[84]. A graphical representation

of the full multivariate model can be found in Supplementary Fig. 6. Model syntax was written following lavaan specifications. Model fitting was done via the lavaan: sem() function (estimator "ML"). In sum, the distinct factor solution provides a multivariate model for the decomposition of phenotypic variances and covariances in genetic and environmental components. Comparison of this model with more parsimonious independent pathway models allows us to test for the presence of a shared genetic (or environmental) component shared across facets.

**Shared-genetic factor solution.** The hybrid-independent genetic pathway model (referred to here as shared-genetic factor solution) is a multivariate approach similar to the correlated factor solution, except with an additional restriction on the genetic covariances between traits ($\sigma_{A,A}$; hence hybrid or genetic, as environmental covariances are modelled in a distinct factor solution fashion). Consider a $5 \times 5$ phenotypic variance-covariance matrix **P**. Under an AE shared-genetic factor solution, **P** can be written as $\mathbf{P} = \mathbf{A_s} + \mathbf{A_u} + \mathbf{E}$, where $\mathbf{A_s} = \mathbf{X_s}\mathbf{X_s}^\mathrm{T}$, with $\mathbf{X_s}$ being a $5 \times 1$ vector of the additive genetic path coefficients of a shared additive genetic factor ($A_C$) loading across all phenotypes, and $\mathbf{A_u}$ is a $5 \times 5$ diagonal matrix including the residual unique genetic variance for each phenotype, $\sigma_{Au}^2$. The full additive genetic variance-covariance matrix can be then as follows:

$$\mathbf{A_t} = \mathbf{X_s}\mathbf{X_s}^T + \mathbf{A_u} = \begin{bmatrix} \lambda_{A1}^2 + \sigma_{Au1}^2 & \lambda_{A1} * \lambda_{A2} & \lambda_{A1} * \lambda_{A3} & \lambda_{A1} * \lambda_{A4} & \lambda_{A1} * \lambda_{A5} \\ \lambda_{A1} * \lambda_{A2} & \lambda_{A2}^2 + \sigma_{Au2}^2 & \lambda_{A2} * \lambda_{A3} & \lambda_{A2} * \lambda_{A4} & \lambda_{A2} * \lambda_{A5} \\ \lambda_{A1} * \lambda_{A3} & \lambda_{A2} * \lambda_{A3} & \lambda_{A3}^2 + \sigma_{Au3}^2 & \lambda_{A3} * \lambda_{A4} & \lambda_{A3} * \lambda_{A5} \\ \lambda_{A1} * \lambda_{A4} & \lambda_{A2} * \lambda_{A4} & \lambda_{A3} * \lambda_{A4} & \lambda_{41}^2 + \sigma_{Au4}^2 & \lambda_{A4} * \lambda_{A5} \\ \lambda_{A1} * \lambda_{A5} & \lambda_{A2} * \lambda_{A5} & \lambda_{A3} * \lambda_{A5} & \lambda_{A4} * \lambda_{A5} & \lambda_{A5}^2 + \sigma_{Au5}^2 \end{bmatrix}$$

(8)

The $5 \times 5$ residual environmental covariance **E** simply contains the unconstrained residual environmental variances and covariances $\sigma_E^2$ and $\sigma_{E,E}$. The $10 \times 10$ twin variance-covariance matrix **M** can then be written as follows:

$$\mathbf{M} = \begin{bmatrix} \mathbf{A_t} + \mathbf{E} & r_\alpha * \mathbf{A_t} \\ r_\alpha * \mathbf{A_t} & \mathbf{A_t} + \mathbf{E} \end{bmatrix}$$

(9)

Where $r_\alpha$ is the expected additive genetic relationship between twins and is fixed to either 1 or 0.5 across MZ and DZ groups, respectively. A graphical representation of the full multivariate model can be found in Supplementary Fig. 9. Model syntax was written in lavaan. Model fitting was done via the lavaan:sem() function. Model comparison between distinct and shared-genetic factor solutions was carried out via the lavaan:: lavTestLRT() function. Here, we additionally note that the shared-genetic factor solution is a less parsimonious version of the more commonly used independent pathway model and, therefore, provides a less restrictive and more specific test for a genetic shared factor when compared to the distinct factor solution.

**Shared-environmental factor solution.** The hybrid-independent environmental pathway model (the shared-environmental factor solution) is a multivariate approach similar to the hybrid-independent genetic pathway model. The $10 \times 10$ matrix **M** can then be written as follows:

$$\mathbf{M} = \begin{bmatrix} \mathbf{A} + \mathbf{E_t} & r_\alpha * \mathbf{A} \\ r_\alpha * \mathbf{A} & \mathbf{A} + \mathbf{E_t} \end{bmatrix}$$

(10)

Where $\mathbf{E_t} = \mathbf{E_s} + \mathbf{E_u}$, with $\mathbf{E_s} = \mathbf{Z_s}\mathbf{Z_s}^\mathrm{T}$, $\mathbf{Z_s}$ being a $5 \times 1$ vector of the environmental path coefficients of a shared environmental ($E_s$) loading across all phenotypes, and $\mathbf{E_u}$ the $5 \times 5$ diagonal matrix including the residual environmental variance for each phenotype, $\sigma_{Eu}^2$.

**Six-variate distinct factor solution.** We extended the previous five-variate distinct factor solution including either music perceptual abilities or general reward sensitivity. This model allowed us to estimate the genetic and environmental correlations between music perceptual abilities or general reward sensitivity and the five facets of music reward sensitivity. To test for the significance of the differences between genetic or environmental overlap between facets, we compared the full six-variate model with a model in which any of the tested pairwise genetic or environmental covariances were set to be equal. Model comparisons were carried out via the laavan:: lavTestLRT() function.

**Structural equation modelling assumptions.** SEM-based estimates obtained from the full-information maximum likelihood (FIML) estimator are unbiased under the assumption that observations follow a multivariate normal distribution[42]. Violation of the assumption of multivariate normality has been found to have little impact on parameter estimates but can have severe consequences for both the $\chi^2$ test statistics and the standard error of the estimates for the parameters. An alternative estimator that is less sensitive or robust to violation of multivariate normality is the maximum likelihood with robust standard error and scaled test statistics (MLR). Although this estimator assumes missingness to be completely at random, it has been shown to provide quite reliable estimates of data missing at random[86]. Relevant comparisons between the two estimators are given in Supplementary Note 1.

### Reporting summary
Further information on research design is available in the Nature Portfolio Reporting Summary linked to this article.

## Data availability
Due to data privacy laws, the raw data from the two waves of data collection, both obtained via the Swedish Twin Registry: Screening Twin Adults Genes and Environment cohort, are protected and, therefore, cannot be made publicly available. Researchers can apply for access to the raw data at the Swedish Twin Registry (see https://ki. se/en/research/research-infrastructure-and-environments/core-facilities-for-research/swedish-twin-registry-core-facility or contact the Swedish Twin registry via email, str-research@meb.ki.se). Applications are reviewed four times a year by a steering committee. To make Figures reproducible, we provided parameter estimates in the Source Data file, including parameter estimates for: the saturated model for music reward sensitivity data partly depicted in Fig. 1a; the twin model of music reward sensitivity data depicted in Fig. 1c, d and Supplementary Fig. 3; the Cholesky decomposition of music perceptual abilities, general reward sensitivity, and music reward sensitivity data depicted in Fig. 2a, b and Supplementary Fig. 5; the distinct factor, shared genetic, and shared environmental solutions for the music reward sensitivity facets scores depicted in Fig. 4 and Supplementary Figs. 6–8; the distinct factor, shared-genetic, and shared environmental solutions for the music reward sensitivity facets scores depicted in Fig. 4 and Supplementary Figs. 6–8; the PC coordinates of music reward sensitivity facet data depicted in Supplementary Fig. 11. The Source Data file also includes extended information for Table 2, namely the observed correlation matrices across the eight phenotypes grouped by the five zygosity groups and twin order, as well as pairwise sample sizes (.np sheets) for complete observations. For simplicity, the Source Data sheet names match the Figures and Table names. Source data are provided with this paper.

## Code availability
Scripts and code used for main analyses can be found at https:// github.com/giacomobignardi/h2_BMRQ and are deposited at https:// zenodo.org/records/14637444[87]. Scripts to apply Latent Variable

Analysis of Twin Data (lavaantwda) using the R package lavaan[83] can be found at: https://github.com/giacomobignardi/h2_BMRQ/tree/main/R/functions/lavaantwda.

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

## Acknowledgements

We thank the Swedish twins for their participation. The Swedish Twin Registry is managed by Karolinska Institutet and receives funding through the Swedish Research Council under grant no 2017-00641. We further thank Aniruddh D. Patel for his critical feedback and Cristina Gonzalez-Liencres for her input on earlier versions of the figures and figures' captions. We also would like to thank the participants of the 2023 Nijmegen *Musicality Genomics Consortium* (https://www.mcg.uva.nl/musicgens/) meeting for insightful discussions that have led to an improved version of the current work. G.B. was supported by the German Federal Ministry of Education and Research (BMBF) and the Max Planck School of Cognition; E.M-H. was supported by a Ramon y Cajal research grant (RYC2020-030748-I) funded by MICIU/AEI/10.13039/501100011033 and by "ESF

Investing in your future"; F.U. was supported by the Bank of Sweden Tercentenary Fund (M11-04512:1), and a donation from Marcus Storch (Dnr 2-6001/2019); G.B., F.U. and S.E.F. were supported by the Max Planck Society.

## Author contributions

G.B. conceived the study, analysed, and visualised the data; G.B. and M.M. drafted the manuscript; F.U., S.E.F. and M.M. supervised the research; L.W.W. validated the work; S.E.F, M.M., R.J.Z., L.W.W. and F.U. conceptually validated the work; G.B., L.W.W., E.M.-H., R.J.Z., F.U, S.E.F. and M.M. revised and reviewed the last version of this manuscript.

## Funding

## Competing interests

The authors declare no competing interests.
