## [Transparent Peer Review file · Nature Communications]

Twin modelling reveals partly distinct genetic pathways to music enjoyment

Corresponding Author: Mr Giacomo Bignardi

Version 0:

Reviewer comments:

Reviewer #1

(Remarks to the Author)

Review by CVDolan of "Distinct genetic pathways to music enjoyment" by Giacomo Bignardi et al

This is a multivariate study based on the classical twin design of 5 facets of the BMRQ, 3 subtests of the SMDT and the BAS-RR subtest. There are 9 phenotypes in total, which are analyzed in three models: the univariate tot BMRQ score model, the trivariate BMRQ total, SMDT total and the BAS-RR model and the 5 facets of the BMQR model. An additional multivariate model is considered (BAS-RR and SMDT predict the 5 facets). This is competent, well-informed work (qua genetic SEM modeling in the classical twin design).

The division of results in main text and suppl is – I guess - dictated by requirement of the journal? It is unappealing that various important info is relegated to the suppl, and to the later methods etc, section.

The hypotheses are

1. To what extent are differences in music reward sensitivity explained by genetic variation?
2. To what extent do genetic effects influence music reward sensitivity above and beyond genetic effects shared with music perceptual ability and general reward sensitivity?
3. To what extent are genetic effects shared between the different facets of music reward sensitivity?

I suppose 1 refers to the total BMRQ score, and 3 refers to the facets. Hypothesis 2 addresses the question concerning unique and common genetic influences on music reward sensitivity, music perceptual ability and general reward sensitivity. Add sentence to explain why this specifically is of interest, specifically why should general reward sensitivity and music reward sensitivity be related? Or discuss music reward sensitivity in more detail with respect to Gray's BIS/BAS scheme. Asking whether genetic effects are unique to music reward sensitivity (above and beyond general reward sensitivity), suggests the hypothesis that music reward sensitivity is a manifestation of general reward sensitivity. Correct?

The hypotheses are formulated exclusively in genetic terms. The possible role of the environment is not of interest? For instance, in Fig 2, the genetic hypotheses correlation =0 and =1 are considered, but not the corresponding environmental hypotheses.

The MBRQ has 5 facets, but also provides a total score. The statistical justification for the use of the total score is based on a phenotypic single factor model (Line 125). Given that the BMRQ is a standard psychometric instrument, I assume that the total score is an accepted and meaningful phenotype. If that is so, why conduct a factor analysis? Is this to prove that in this dataset (this target population), the 5 facets are unidimensional? If so, why is that necessary? Interestingly, it emerges later that the A covariance matrix (of the facets) is not unidimensional. That renders the actual meaning of the total score hard to interpret. Whether or not the E covariance matrix is not considered as the authors are largely uninterested in environmental sources of individual differences.

By Line 116 (results), we dive into the results, the five facets of the BMRQ are mentioned, but with little explanation. A clearer explanation / description of the actual phenotypes could be welcome. E.g., I associate musicality with "the fact or quality of creating, performing, or interpreting music in a highly skilled and artistic way", but it is used more broadly here?

L117. The sample and BMRQ descriptives. There are 5 + 3 + 1 phenotypes, but descriptives are limited to the BMRQ total score. E.g. table of twin correlations (5 groups, 9 phenotypes), phenotypic correlations among the 9 phenotypes, marginal descriptives, etc.

From Table 1, it seems that there is a lot of missingness. How come? Was missingness tested for MCAR?

L 143 There is a main effect of sex on the total BMRQ scores. The main effect of sex on the BMRQ total score is based on the LRT, i.e., $\chi^2(30)=300$. Whence $df=30$, given that the main effect of sex is tested in a univariate SEM? For the main effect of sex, it would be useful to report a sensible effect size, i.e., Cohen's D.

L 144 Main age and sex effects are tested in the SEM, I assume. Hence the variation in approach (sex main tested using LRT but age results in terms of 95CI)? The effect of age is small $\beta_{age} = -.03$. Is β_{age} the standardized regression coefficient? What is the R^2 attributable to age?

L159. The comparison of the ADE model for total BMRQ score and the "baseline model" yields a χ^2 of 41.13 with 33 degrees of freedom. Whence $df=33$ if this is a univariate twin model?

Line 195 "Fig. 2. Schematic illustration of the sequential decomposition approach." This is a.k.a. a Cholesky to triangular decomposition, but we apparently need a new terms. Don't the three models (fig 2 A, B, C) simply correspond to the hypotheses that the genetic correlation r is $r=0$ (A) or $r=1$ (C) (B, $0 < r < 1$)? That could be pointed out.

"we use Structural Equation Modeling (SEM), informed by the Classical Twin Design 140 (CTD). First, as a baseline for further model comparisons, we fit a univariate model to 141 individuals' BMRQ total scores (Fig. 1A; age and sex were accounted for)."

Check tenses. We used.... We fitted. Past tense, please.

Line 63 "Despite the widespread power of music, however, it should also be noted that many people do not occupy themselves with music". And yet in the abstract: "the biology of a key aspect of human behaviour." If many people do not occupy themselves with music, "key" is hyperbolic?

"First, as a baseline for further model comparisons, we fit a univariate model to (Line141) individuals' BMRQ total scores" The statement that the baseline model is a univariate model is not informative. The baseline model is the saturated model?

Suppl table 1 includes tests "Quantitative", "Qualitative". These are not explained. In the main text I could not find any reference to these. Generally, the suppl appears to contain information, which is poorly referenced in the main text.

L 226 "note that the environmental path from music perceptual abilities to music reward sensitivity is not significant, indicating only common genetic causes. Between parentheses, the significant path from the E component to general reward sensitivity ($p = .03$) ..."

Here are throughout this paper. Statistical tests are conducted, and p values are reported, implying statistical testing. Statistical tests should include an explicit statement concerning alpha. I assume alpha is .05. Is it .05 everywhere? Are the actual number of statistical tests a consideration in deciding on the appropriate alpha?

Figure 3 depicts various results: the phenotypic correlations, the Cholesky decomposition results, and the implied decomposition of phenotypic variance. The parameter estimates of fig 3B are supposed to be given in Suppl Table 3. 1. The parameter estimates are not all visible in Fig 3B. So that is inconvenient. I suppose that the discrepancy between the fig 3 b results and the suppl table 3 results is explained somewhere.

Figure 4 depicts various hypotheses concerning the relationship between facets... Are these the 5 facets of the BMRQ (I assume, this could be stated explicitly). The representation as path diagrams is hard to follow. 4A why is the orange colored factor call a common factor given that it only varies on 1 on the 5 facets / phenotypes? What does the "=" signify in 4A? I assume that 4B is a correlated factors model. The pathdiagrammatic representation is not correct (given path diagram conventions). Is the about the 5 BMRQ facet? Why are there 4 variables? It is strange that in 4A the hypothesis of uncorrelatedness extends to the E influences. Why would the hypothesis concerning A (Genetic) influences extend to the E influences (in 4A and 4B, but not in 4C). The explanation of 4C is hard to follow: "The common latent genetic factor (in orange) could 261 explain all the genetic variance associated with one facet (e.g., dashed circle)." The dashed circle does not pertain to the all the genetic variance, it pertains to the residual variance. If the dashed latent variable has zero variance, the correlation between the associated phenotype and the common factor is 1. That would imply that the common factor and the observed variable are indistinguishable (meaning that we do not really need the common factor at all). All this is about modeling the associations between the 5 facets, but are the facet correlations actually reported? Why is reference to these "Phenotypic correlations can be found in Supplementary Fig. 4" in the figure caption of fig 5. The correlations are the focus, but are hardly mentioned.

L 234 The results concerning the 5 facets are based on an AE model. Are the twin correlations reported? Somewhere in the suppl perhaps. This twin correlations are important info, really. Note that the choice for an AE model implies that an ACE or ADE is not suitable? Was that established?

Results in Fig 5 are given in terms of h^2 and A and E correlations. The reader may ask: how much does A contribute to the phenotypic covariance of (say) emotion evocation and mood reg? That question is not simple to answer based on this info. A genetic correlation is obviously informative (e.g. pleiotropy) as is h^2 (individual differences / variance), but is not the question here: what do A and E contribute to the phenotypic covariance? The answer, I think, is 50% vs 50%, notwithstanding the A correlation of .74.

The path diagram 6A does not include correlated residuals among the dependent variables (5 facets). Is that correct? Does this path diagram correspond to the hypothesis of interest? Namely uncorrelated residuals? Why is it that now the E model mirrors the A model (in contrast to the previous). Why does 6A include the unit vector (triangle) where as most of the other path diagrams do not? Note that in the text various statistical test outcomes are mentioned. As usual here without explicit reference to alpha or alpha given multiple testing (beyond the $<.05$, which suggests that the alpha is .05 throughout as applied to every single statistical test ... how many tests? What is the policy w.r.t. multiple testing).

318 Given the lack of genetic dimensionality (5 facets do not fit a common factor model), this must mean that the phenotypic structure is not unidimensional (even though the E structure could be unidimensional – although this is apparently not of interest here). So, what light does that shed on the analyses of the facet total score, and on the results: L 124 “A confirmatory factor model showed acceptable fit for a model with a single latent music reward sensitivity factor capturing correlations between the five facets (CFI = .96, SRMR = .035).” And on the “notably high heritability” of the BMRQ total score? The authors note correctly: “Our results thus may challenge the epistemological status of music reward sensitivity as a latent causal factor (43,55,56, as a latent factor is unlikely to hold unless a common-genetic factor solution holds (for additional conditions, see 43).” Here again the focus on A structure. I suppose that the additional conditions concern the E structure? And the common pathway model. Again, it is unclear why the E structure (and E in general) is discarded.

319 “Instead, these findings are consistent with musical enjoyment being built upon genetically interconnected yet partly distinct parts.”

This should be “individual differences in musical enjoyment”

326 “Yet, the finding of notably high heritability for music

327 reward sensitivity gives hope for molecular genetic studies to answer questions about genetic

328 underpinnings of musicality in general and musical affect in particular.”

Notably high heritability is .54 (in the AE model)? Why is that “notably” high? Apparent the e^2 ($\text{var}(E) / \text{var}(ph)$) is also “notably high” (.46)? I suppose that .54 would be notably high if we had expected low h^2 , but no such expectation was expressed.

Whence this emphasis on passive gene-environment correlation? The participants are adults, the absence of C in adults does not imply an absence of C in children, where passive gene-env correlation may be relevant. The discussion is about passive cov(AC). Why would that apply in adults? Btw: musical talent, and the positive feedback loop between genetic disposition and env circumstances is a primary example of active A-E covariances due to niche-picking (but this may apply to music production in proficient musicians).

417 “An additional assumption is the lack of gene-by-shared environment interaction,

418 which could lead to an underestimation of the variance of the C component.” AxC unmodelled results in overestimation of A variance? See L 562 “If AxC is present, then σ^2

% 563 “ $> \sigma^2_A$ is expected”

(Remarks on code availability)

Reviewer #2

(Remarks to the Author)

This was a very interesting article reporting on a genetic decomposition of music reward and its association with music perception and general reward sensitivity. The article is important as there have not been other genetic investigations on music reward sensitivity and how they relate to other constructs. I found the manuscript very well-written and appreciated the authors' succinct notes on where model assumptions were made and what biases would be introduced if these assumptions were violated. I therefore struggled to find places to give critical feedback. Below I note some comments, but please note that I consider almost all these comments minor. My most substantial concern is that I think some parallel questions/analyses should address environmental influences in addition to genetic influences (see below).

Introduction:

This section was well-written and clear. My only concern is that the article focuses primarily on genetic influences, but the nature of the twin dataset is also informative to the environmental influences on music reward and its overlap with other traits. I think unpacking these genetic associations is very interesting and important, but I suggest expanding the key research questions (e.g., lines 97-101) to include genetic and environmental variation, as these are two sides of the same coin.

Results:

It's unclear why the authors state that comparison of within-pair MZ and DZ correlations are an estimate of narrow-sense heritability. I would argue that this is an estimate of broad sense heritability, especially since they state that $r_{MZ} > r_{DZ}$ which indicates evidence for both additive and nonadditive genetic components (i.e., broad-sense heritability).

I like that Figure 1 displays both the results for the ADE and AE solutions of the initial model. However, I would use the phrase 'more parsimonious' rather than 'better fit' when referring to the selection of the AE model over the ADE model. The AE model did not fit significantly worse than the ADE model, but it also does not fit better (e.g., it has the same AIC and BIC values), it is just a more parsimonious model.

Either in the Figure 2 caption or in lines 187-191 it would be good to mention that these three possible outcomes could be observed for environmental influences as well. I don't think there is reason to expect environmental influences will only be 'partial separation' as depicted (Note: it makes sense for the figure itself to emphasize the genetic differences, I just suggest adding a note that these possibilities are displayed for genetic influences but apply to environmental influences as well).

I can be convinced to keep Figure 2B as is, but it may help readers unfamiliar with genetic analyses to present a standardized version of this figure in the main text. For example, it makes sense that the main text and Figure 2C decompose variance in music reward sensitivity, but it's hard to tell what the heritability and genetic/environmental covariance for music perceptual abilities and general reward sensitivity are from this figure without doing a lot of computations by hand.

Given the emphasis on model selection in Figure 2, it would help to show specific tests to determine that the trivariate genetic model supports partial separation vs. complete separation of genetic or environmental influences on music reward vs. other phenotypes (e.g., chi-square test when genetic/environmental cross-paths are removed). Again, I think this is important to report for both genetic and environmental influences.

In lines 264-271, please report whether the environmental influences also support a distinct factor solution (like genetic influences). Also, it was hard to tell from Supplementary Table 4 whether distinct vs. common factors were tested for genetic and environmental influences separately (which it probably should be), or if the common factor model that was rejected collapsed both A and E influences at the same time.

Discussion:

If you agree with my earlier suggestions, you will need to comment on whether the nonshared environmental influences show similar patterns as the genetic results in the discussion. Even if they don't, these findings would still be relevant to our understanding of music reward and related constructs. Note: the lack of shared environmental influences is addressed in a compelling way here, but I think some discussion should also talk about nonshared environmental influences as well.

It would be useful to comment briefly in the limitations paragraph on how the 10 year gap between administration of the SMDT and reward measures may impact the results. I suspect this would underestimate phenotypic and genetic/environmental correlations with SMDT.

(Remarks on code availability)

Reviewer #3

(Remarks to the Author)

I appreciated reading this interesting study, which uses behavioural data from a large sample of Swedish twins to investigate genetic factors underlying music enjoyment. On a general level, its interdisciplinarity, breadth, rigour, and potential scientific and popular impact make it appear well-suited for eventual publication in Nature Communications. I do not feel confident enough in the specific details of the methods to make strong comments about them, but from what I can tell they mostly seem to be reasonable methods and their conclusions largely seem reasonable. So I mainly only have relatively minor suggestions that I think might help clarify the study and better highlight its strengths and limitations:

Restricted sample: The main limitation I see is that the sample is not only restricted to twins, but SWEDISH twins. I was expecting to see some discussion of this after the discussion on limitations of twin studies, so was surprised to see it not mentioned at all, as if we could assume that genetic and environmental factors underlying musical reward in Swedish twins could be assumed to generalize to all humans throughout the world. The issue of non-representativeness of samples from Europe and North America is increasingly recognised as one of the key challenges of music cognition (Jacoby et al., 2020) and the behavioural sciences more generally (Henrich et al., 2010), so certainly seems to deserve more acknowledgment.

Title: I find the title, "Distinct genetic pathways to music enjoyment" overly vague. Best practice would be to give at least a hint of the data/methods ([<https://www.nature.com/articles/s41562-023-01596-8>])(<https://www.nature.com/articles/s41562-023-01596-8>). In particular, I think you really should include "Swedish twins" somewhere in the title (see above), and perhaps an indication of the BMRQ or more generally that you are analysing behavioral data. For example, "Behavioral data from Swedish twins reveals distinct genetic pathways to music enjoyment".

I should mention here that the term "pathway" also feels a bit vague - to me it implies a clearer mechanism linking genotype

and phenotype than I see here. However I'm not confident I can necessarily offer a better term - "factor" came to mind, but I'm not sure it's quite what is needed. In any case, I suggest considering the term and perhaps defining it more clearly if you intend to continue using it so prominently (and maybe also avoiding using confusingly similar terms like "the auditory cortex and its pathways...")

Abstract:

"genetic factors substantially explain variance in music reward sensitivity above and beyond genetic influences shared with music perception and general reward sensitivity" - I think this would benefit from a quantitative indication here - for example, the relative size of genetic and environmental effects?

"large sample of Swedish twins (N = 9,169)" - this is good to mention this here, but isn't "N = 9,169" a little misleading if this only included 2305 pairs as suggested in Table 1? Would something like "N = 4,610 individuals (2,305 pairs)" be more accurate? Apologies if I've misunderstood the methods.

"behavioural data on several facets of music reward sensitivity, music perceptual ability, and general reward sensitivity" - might be good to specify that the "primary measure" is the BMRQ.

If you have room, I suggest it might be worth mentioning the intriguing result that "music perception shows stronger genetic correlations with social bonding than other facets of music reward" as you discuss later in the paper.

Novelty of data:

It wasn't obvious to me how much of the analyses reported analysed new data and how much was reanalysis of existing data. For example: "a large sample of deeply phenotyped monozygotic (MZ) and dizygotic (DZ) twins with available musicality data" - where do this "available musicality data" and "large sample" come from? Ref. 13 is cited at one point but it's not immediately clear to me how much (if any) of the data reported was previously published in ref. 13. I think it should be clear to a reader of this article what is new data and what is previously published data without having to read external references. This is particularly important since the current data are not made publicly available ("as registry data were used", which sounds reasonable), and so cannot be independently checked.

Figures:

In general, I found the figures a bit confusing. The main problem is that they were peppered with acronyms that were difficult to interpret (e.g., "E mrs", "DZos", "ee", etc.). I suggest trying to re-draw some to write out key acronyms in full in some places, remove them if not essential, and otherwise try to at least define them more clearly in the caption.

In particular, the letters "A" and "E" appear to have much importance, but I can't tell what they represent. Does "A" mean "additive genetic" and "E" mean "environmental" (inferred from the Fig. 5 caption)? If so please make that clearer!

3b: white text boxes seem to be obscuring some lines

Discussion:

Given the close relationships between music and language (e.g., Nayak et al., 2022; Patel, 2008), I would have liked to see some speculation about whether we expect the "pathways" to be distinct or shared with language (e.g., reward from conversation).

"genetic variance associated with music reward sensitivity, beyond perceptual and general reward processing" - what is the mechanism here?

"different aspects of musicality, namely producing music and enjoying music, might follow different patterns of intergenerational transmission" - seems to imply that "intergenerational transmission" of music(al)ity is only via genes but of course cultural transmission is probably equally or more important. Better to use different words like "genetic inheritance" here if that's what you mean.

"Anirudh" in Acknowledgments missing an extra "d"

"Exploratory analyses" - typically refer to non-preregistered analyses, contrasted with "confirmatory" analyses that are pre-registered. But here no analyses are pre-registered, are they? Regardless, I think Nature Communications requires statements about whether analyses are pre-registered, which I don't see in the current manuscript.

I'm sure there are other ways the manuscript could be improved, but I hope these suggestions will at least be helpful.

Signed,
Patrick Savage

References:

Henrich, J., Heine, S. J., & Norenzayan, A. (2010). The weirdest people in the world? *Behavioral and Brain Sciences*, *33*, 61–135. (<https://doi.org/10.1017/S0140525X0999152X>)

Jacoby, N., Margulis, E. H., Clayton, M., Hannon, E., Honing, H., Iversen, J., Klein, T. R., Mehr, S. A., Pearson, L., Peretz, I., Perlman, M., Polak, R., Ravnani, A., Savage, P. E., Steingo, G., Stevens, C., Trainor, L., Trehub, S., Veal, M., & Wald-Fuhrmann, M. (2020). Cross-cultural work in music cognition: Methodologies, pitfalls, and practices. *Music Perception*, *37*(3), 185–195. (<https://doi.org/10.1525/mp.2020.37.3.185>)

Nayak, S., Coleman, P. L., Ladányi, E., Nitin, R., Gustavson, D. E., Fisher, S. E., Magne, C. L., & Gordon, R. L. (2022). The Musical Abilities, Pleiotropy, Language, and Environment (MAPLE) framework for understanding musicality-language links across the lifespan. *Neurobiology of Language*, *3*(4), 615–664. (https://doi.org/10.1162/nol_a_00079)

Patel, A. D. (2008). *Music, language and the brain*. Oxford University Press. (<https://doi.org/10.1093/acprof:oso/9780195123753.001.0001>)

(Remarks on code availability)

Version 1:

Reviewer comments:

Reviewer #1

(Remarks to the Author)

Twin modelling reveals partly distinct genetic pathways to music enjoyment (revision).
The authors have acted adequately on all my previous remarks.

Here are some residual remarks.

Figure 1 C header is incorrect. These are not DZ twins, they are MZ twins.

“Footnote: Circles are the latent component; 199 Double-headed arrows within circles...”

“Circles are the latent component”. Does not make sense because component is singular. Also the term component here is ambiguous: Circles represent latent variables. The double headed arrow is not “within circles”, Some connect different variable, other connect the variable with itself. Within the circle implies that the double headed arrow is “within” the circle.

I do not understand the notation “Squares|Rectangles”

“the triangle, the phenotypic mean grouped by twin order201 (panel a) and sex (panel c) already adjusted for age;”

The triangle represents the unit vector not the phenotypic mean. The regression of the phenotype on the unit vector is standard in linear regression to estimate the intercept, which coincides with the phenotype mean here. “by twin order” refers to what ordering? how are they ordered.

223. “Removing the genetic cross-paths from music223 perceptual abilities and general reward sensitivity to music reward sensitivity, respectively,224 worsened the model fit ($\chi^2(1)\Delta df = 94.37$ and $\chi^2(1)\Delta df = 87.71$, both $p < .001$), indicating partial225 overlap between genetic factors (see Supplementary Table 5).”

Given the fact that the readership is not necessarily familiar with the Cholesky parameterization of a covariance matrix, but would be useful to add a sentence. E.g., concerning $\chi^2(1)\Delta df = 94.37$. What is this a test of, not formulated in terms of the Cholesky. E.g., say, “This is a test of the hypothesis that the correlation between the additive genetic variable of music perceptual abilities and the additive genetic variable of music reward sensitivity is zero”.

Btw: the notation is weird. Should this be $\chi^2(\Delta df=1) = 94.37$?

I assume that the authors like the Cholesky because of the “sequential nature” (in the linear regression sense), but ultimately the representation of the completely standardized AE model with correlations among the three additive genetic variables and the three E variables, and $\sqrt{h^2}$ and $\sqrt{1-h^2}$ or $\sqrt{e^2}$ is easier to understand.

Figure 4 “b Genetic and environmental effects315 on music reward sensitivity are partially heterogeneous.” What does heterogeneous mean here?

“The likely absence464 of shared environmental effects may imply only a small, if present, passive gene-environment465 correlation (e.g., genotypes associated with music reward sensitivity in the parents influence466 the children via the environment the parents provide and the genes they pass on to their467 children, see ref.60).”

The result was obtained in adult. The absence of C does not imply anything about cov(AC) in children. E.g., there is

influence of C and covAC in children's IQ. But there is not C in adults IQ. How do the results obtained in adults generalize to possible interplay in children? See "The lack of common491 environmental effects in adulthood, as found in this study, does not imply a lack of common492 environmental effects on similar musicality traits earlier in the lifespan."

"On the contrary, this is to be expected if evocative and active gene-environment484 correlations are at play, which seems likely for traits related to music enjoyment. Such gene-485 environment correlations would not inflate heritability estimates."

But Purcell (2002 Twin Research Volume 5 Number 6 pp. 554± 571) demonstrates that
AxE inflates E variance in the CTD
AxC inflates A variance in the CTD
+rAC inflate C variance in the CTD
+rAE inflates A variance in the CTD

So +rAC and +rAE inflate C and A variance respectively but the heritability (standardized A variance) is not inflated?

649 "The latter case would instead result in active gene-environment correlations that650 are still consistent with the estimate of the variance components."

gene-environment correlations are consistent with the estimate of variance components. This does not make sense. Also note that "the estimate" is singular and components is plural.

(Remarks on code availability)

Reviewer #2

(Remarks to the Author)

The authors have addressed all my concerns and they were very responsive to those concerns of the other reviewers. I believe this manuscript is suitable for publication

(Remarks on code availability)

Reviewer #3

(Remarks to the Author)

I thank the authors for their comprehensive response, which has resolved almost all of my concerns.

I'm still not 100% convinced by the justification of leaving out "Swedish" from the title, given the (good) new discussion about how we can't necessarily generalise these results outside of Northern Europe. However, I'm happy to leave the final decision on this to the authors/editors.

Signed,
Patrick Savage

(Remarks on code availability)

Point-by-point response

Dear Reviewers,

We are grateful for the thorough review and excellent feedback from all referees. We have revised our paper based on the comments and answered each of them below. Please see the original review *comments in italics*, followed by our response in **blue**. Major changes in the manuscript are highlighted in **red** and reported verbatim below. We believe that the changes implemented following the advice of the reviewers resulted in a substantially improved manuscript.

Giacomo Bignardi,
On behalf of the Coauthors.

Reviewer #1 (Remarks to the Author):

Review by CVDolan of “Distinct genetic pathways to music enjoyment” by Giacomo Bignardi et al

This is a multivariate study based on the classical twin design of 5 facets of the BMRQ, 3 subtests of the SMDT and the BAS-RR subtest. There are 9 phenotypes in total, which are analyzed in three models: the univariate tot BMRQ score model, the trivariate BMRQ total, SMDT total and the BAS-RR model and the 5 facets of the BMQR model. An additional multivariate model is considered (BAS-RR and SMDT predict the 5 facets). This is competent, well-informed work (qua genetic SEM modeling in the classical twin design).

We thank Reviewer 1 for the thorough review. Below, we highlight how we addressed all points raised.

Comment #1

The division of results in main text and suppl is – I guess - dictated by requirement of the journal? It is unappealing that various important info is relegated to the suppl, and to the later methods etc, section.

We have followed Nature Communications guidelines for reporting results (<https://www.nature.com/ncomms/submit/article>). In the updated version, following the Reviewer's suggestion, we have re-introduced some information that was previously relegated to the Supplement only. Changes are outlined in the point-by-point response letter.

Comment #2

The hypotheses are

- 1. To what extent are differences in music reward sensitivity explained by genetic variation?*
- 2. To what extent do genetic effects influence music reward sensitivity above and beyond genetic effects shared with music perceptual ability and general reward sensitivity?*
- 3. To what extent are genetic effects shared between the different facets of music reward sensitivity?*

I suppose 1 refers to the total BMRQ score, and 3 refers to the facets. Hypothesis 2 addresses the question concerning unique and common genetic influences on music reward sensitivity, music perceptual ability and general reward sensitivity. Add sentence to explain why this specifically is of interest, specifically why should general reward sensitivity and music reward sensitivity be related? Or discuss music reward sensitivity in more detail with respect to Gray's BIS/BAS scheme

Question 1 indeed refers to the total BMRQ sum score, and question 3 refers to the facets. We have now clarified this in the text. Regarding the final point, a theoretically interesting question, which stimulated this study, is whether differences in sensitivity to music rewards simply reflect differences in general reward sensitivity, or if music reward sensitivity on the contrary is – at least in part – distinct, with separate genetic and environmental influences.

To make this point clearer, we added the following section before introducing our hypothesis (lines 101-106):

When studying sources of individual differences in music reward sensitivity, it is also crucial to consider variation in more general perceptual or hedonic processes, as the former can be merely a consequence of the latter (e.g., a specific lack of pleasure derived from music could be one aspect of a general lack of sensitivity to rewarding stimuli). As such, we complemented our study by including measures of music perceptual abilities and general reward sensitivity, and addressed the following three research questions:

Comment #3

Asking whether genetic effects are unique to music reward sensitivity (above and beyond general reward sensitivity), suggests the hypothesis that music reward sensitivity is a manifestation of general reward sensitivity. Correct?

We wanted to test if there are distinct genetic and environmental influences on music reward sensitivity, i.e., that it is not simply a manifestation of general reward sensitivity (see also comment #2 above). Our results are consistent with this expectation.

Comment #4

The hypotheses are formulated exclusively in genetic terms. The possible role of the environment is not of interest? For instance, in Fig 2, the genetic hypotheses correlation =0 and =1 are considered, but not the corresponding environmental hypotheses.

While this study mainly focused on the genetic sources of differences in music reward sensitivity, we appreciate the reviewers' interest in the role of the environment (see also Reviewer 2 comments). Therefore, we have complemented our analysis, results, Figures, and discussion throughout the manuscript to reflect both genetic and environmental influences on music reward sensitivity. In the introduction, we now mention (lines 125-127):

All analyses were complemented by an investigation of environmental influences on music reward sensitivity and their overlap with environmental effects on music perceptual ability and general reward sensitivity.

Comment #5

The MBRQ has 5 facets, but also provides a total score. The statistical justification for the use of the total score is based on a phenotypic single factor model (Line 125). Given that the BMRQ is a standard psychometric instrument, I assume that the total score is an accepted and meaningful phenotype. If that is so, why conduct a factor analysis? Is this to prove that in this dataset (this target population), the 5 facets are unidimensional? If so, why is that necessary? Interestingly, it emerges later that the A covariance matrix (of the facets) is not unidimensional. That renders the actual meaning of the total score hard to interpret. Whether or not the E covariance matrix is not considered as the authors are largely uninterested in environmental sources of individual differences.

Given that the BMRQ had not previously been validated in Swedish, we conducted a factor analysis on the BMRQ facets to confirm whether the Swedish version provides a reliable unidimensional measure of music reward sensitivity. This analysis ensures that the total score is statistically justifiable and relevant for phenotypic analyses in this population.

Notably, the use of a total score at the phenotypic level does not imply that the underlying genetic or environmental covariance matrices exhibit a unidimensional structure. In fact, our results indicate that while the facets may be aggregated into a phenotypic total score, each facet reflects distinct genetic influences, highlighting nuanced genetic contributions to different aspects of music reward sensitivity.

Comment #6

By Line 116 (results), we dive into the results, the five facets of the BMRQ are mentioned, but with little explanation. A clearer explanation / description of the actual phenotypes could be welcome. E.g., I associate musicality with “the fact or quality of creating, performing, or interpreting music in a highly skilled and artistic way”, but it is used more broadly here?

We indeed use “musicality” in a somewhat broader sense, i.e. referring to the capacity to perceive, produce, and enjoy music; therefore, it is not exclusive to highly skilled individuals. This is now explained more clearly in the manuscript. Moreover, in the introduction, we added a section describing the different phenotypes studied (lines 92-97):

...a total score to assess how much individuals derive pleasure from music, as well as fine-grained characterisation of individual differences in emotion evocation (emotional reactions to music), mood regulation (the impact of music on individuals’ moods), music seeking (the tendency to seek out new music), sensory-motor (deriving pleasure from movements evoked by music), and social reward (deriving pleasure from social bonding through music) facets of music enjoyment

Comment #7

L117. The sample and BMRQ descriptives. There are 5 + 3 + 1 phenotypes, but descriptives are limited to the BMRQ total score. E.g. table of twin correlations (5 groups, 9 phenotypes), phenotypic correlations among the 9 phenotypes, marginal descriptives, etc.

We thank the Reviewer for noticing this omission. We have now included an additional Supplementary Table (Supplementary Table 1), with descriptives for the additional eight phenotypes (means and *SD*). (in order: 1 BMRQ total score + 1 SMDT and 1 BASrr total score + 5 facets scores of BMRQ.) Furthermore, we have provided the Supplementary Data 3 (https://github.com/giacomobignardi/h2_BMRQ/blob/main/SFILE/04_supplementary_data_3.xlsx) with the observed correlation matrices, stratified by the five zygosity groups across the eight phenotypes.

Comment #8

From Table 1, it seems that there is a lot of missingness. How come? Was missingness tested for MCAR?

The response rate is low because our study is population-based (i.e., we have invited a whole birth cohort to participate in the Humans Making Music studies). We have made this more explicit in the manuscript (lines 545-548).

We note that response rates in the STAGE cohort have been low (~30%). The low response rates presumably reflect that this is a population-based study (i.e., where a whole birth cohort of Swedish twins is invited to participate) of a working-aged cohort; this is a general phenomenon not unique to our study.

Furthermore, we found no evidence for sampling bias effects on the BMRQ (See Supplementary Table 2), consistent with data missing at random. More precisely, we ran a series of analyses to test whether the missingness of one twin from a pair is related to the measure of interest (Neale & Eaves, 1993). The results show that the BMRQ scores of single twins (singletons) were not significantly different from the BMRQ scores of twins in complete pairs. However, SMDT scores of single twins were significantly lower than the SMDT scores of the complete pairs of twins, suggesting some sampling bias with regard to music perceptual abilities. We have added this finding to the discussion and highlighted it as an issue for future studies (lines 506-512).

...although we found no evidence of sampling bias on music reward sensitivity, we also found evidence that individuals with lower-than-average music perceptual abilities tend to be slightly underrepresented in our study (see Supplementary Table 2). This finding is not completely surprising, as population-based studies based on voluntary participation, such as ours, are known to be subject to participation bias. Nevertheless, the effect was small and evident only for a secondary measure in this study.

Comment #9

L 143 There is a main effect of sex on the total BMRQ scores. The main effect of sex on the BMRQ total score is based on the LRT, i.e., $\chi^2(30)=300$. Whence $df=30$, given that the main effect of sex is tested in a univariate SEM? For the main effect of sex, it would be useful to report a sensible effect size, i.e., Cohen's D.

The difference in degrees of freedom (df) between the saturated model and the model with constrained means across sex equals 30 because the latter model contains several constraints.

As can be seen in Supplementary Table 1, the "Same mean" model is recursively nested in the "Baseline covariate" model (Saturated model but with variance in the manifest covariate, that is, age, constrained to be equal across groups, 1 parameter instead of 5, 4 df); "Twin order" model ("Baseline covariate" model but with covariate effects constrained to be equal and with means and variances in twins 1 and 2 constrained to be equal across groups, 1 parameter instead of 10 for beta age, 12 parameters instead of 20 for birth order, $17 + 4 = 21$ df); "Zygosity" model ("Twin order" model but with zygosity-specific means and variances constrained to be equal across groups, 4 parameters instead of 12, $8+17+4 = 29$ df). As such,

the “Same mean” model is 30 *df*, as the means across sexes are constrained to be equal (1 parameter instead of 2, $1 + 29 = 30$ *df*). We have now referenced the Supplementary material where appropriate, highlighted this in Supplementary Table 3, and provided *p*-values related to the LRT against the saturated model only to avoid further confusion.

Further, as suggested, we have now reported Cohen’s *d* ($d = 0.37$) for the difference between the mean scores. We have calculated Cohen’s *d* as $(M_w - M_m)/\sigma$, since σ was constrained to be equal across sexes. Note that an alternative *d* (reported in Supplementary Table 1, for completeness) obtained from a model with unconstrained variances $(M_w - M_m)/\sqrt{(\sigma_w^2 + \sigma_m^2)/2}$ resulted in a virtually identical value, $d = 0.37$.

Comment #10

L 144 Main age and sex effects are tested in the SEM, I assume. Hence the variation in approach (sex main tested using LRT but age results in terms of 95CI)? The effect of age is small $\beta_{age} = -.03$. Is β_{age} the standardized regression coefficient? What is the R^2 attributable to age?

The main effect of age was indeed tested in the SEM. Previously, we reported the standardised effect of age (so R^2 is 0.1%). However, given the inconsistency between the reported effects (95% CI and LRT χ^2), we now harmonised the report, reported the unstandardized estimate, the percentage of variance explained, and applied LRT to increase clarity (lines 168-169).

$\beta_{age} = -.06$ (95% CI [-0.10,-0.02]; $\sigma_{age} = 57.90$ years, $\chi^2(1)_{\Delta df} = 8.58$, $p = .003$), explaining only 0.1% of the total BMRQ variance

Comment #11

L159. The comparison of the ADE model for total BMRQ score and the “baseline model” yields a χ^2 of 41.13 with 33 degrees of freedom. Whence $df=33$ if this is a univariate twin model?

Apologies for the confusion, and thanks again for providing the opportunity to clarify. We have reported the comparison with the saturated model. The ADE model includes the same constraints as the “Baseline covariate”, “Twin order”, “Zygosity”, and “Sex” (“Same variance, Same covariances (MZ and DZ)”, and “Same covariances (DZ and DZos)”) models, as explained above. Therefore, compared to the saturated model, we have the previous $4 + 17 + 8$, plus 1 (“Same variance”) + 2 (“Same covariance DZ and MZ”) + 1 (“Same covariance DZ and Dos”). Hence, the 33 *df*.

Comment #12

Line 195 *"Fig. 2. Schematic illustration of the sequential decomposition approach." This is a.k.a. a Cholesky to triangular decomposition, but we apparently need a new terms. Don't the three models (fig 2 A, B, C) simply correspond to the hypotheses that the genetic correlation r is $r=0$ (A) or $r=1$ (C) (B, $0 < r < 1$)? That could be pointed out.*

Yes, the three models correspond to $r=0$ (Fig. 2a), $0 < r < 1$ (Fig. 2b), and $r=1$ (Fig. 2c). However, we have now decided to remove Fig. 2 as we realise it may confuse the reader. Instead, we now simply describe our approach in the text while keeping the general readership of Nature Communications in mind.

Regarding the term 'sequential', we have decided to accommodate the Reviewer 1 suggestion and switched to a classical Cholesky decomposition. Textual changes can be found in the main text. Please note that, in response to a request from Reviewer 2, we now report estimates for the Cholesky decomposition of standardised variables; see now Fig. 2 (previously Fig. 3).

Comment #13

"we use Structural Equation Modeling (SEM), informed by the Classical Twin Design 140 (CTD). First, as a baseline for further model comparisons, we fit a univariate model to 141 individuals' BMRQ total scores (Fig. 1A; age and sex were accounted for)."

Check tenses. We used.... We fitted. Past tense, please.

Many thanks. We have amended the tenses.

Comment #14

Line 63 *"Despite the widespread power of music, however, it should also be noted that many people do not occupy themselves with music". And yet in the abstract: "the biology of a key aspect of human behaviour." If many people do not occupy themselves with music, "key" is hyperbolic?*

Good point. We have amended the highlighted sentence in the text and have tempered our statement in the abstract by changing the word "key" to "important".

Comment #15

"First, as a baseline for further model comparisons, we fit a univariate model to (Line141) individuals' BMRQ total scores" The statement that the baseline model is a

univariate model is not informative. The baseline model is the saturated model?

We thank the reviewer for pointing this out. For simplicity, we now report comparisons with the saturated model, since the “baseline covariate model” is not a saturated model. The baseline covariate model is a constrained version of the saturated model, with age variances across groups constrained to be equal. Note that we have changed the name “baseline” to “baseline covariate” to avoid further confusion.

Comment #16

Suppl table 1 includes tests “Quantitative”, “Qualitative”. These are not explained. In the main text I could not find any reference to these. Generally, the suppl appears to contains information, which is poorly referenced in the main text.

We thank the Reviewer for spotting this omission. We have now reframed the models as “Same covariances (MZ and DZ)” and “Same covariances (DZ and DZos)” and refer to the necessary information in Supplementary Table 3. We note that we have opted to report only the main results that are strictly pertinent to our primary aims in the main paper and relegate other findings (e.g. sex differences) to the Supplementary Material.

Comment #17

L 226 “note that the environmental path from music perceptual abilities to music reward sensitivity is not significant, indicating only common genetic causes. Between parentheses, the significant path from the E component to general reward sensitivity ($p = .03$) ...” Here are throughout this paper. Statistical tests are conducted, and p values are reported, implying statistical testing. Statistical tests should include an explicit statement concerning alpha. I assume alpha is .05. Is it .05 everywhere? Are the actual number of statistical tests a consideration in deciding on the appropriate alpha?

We apologise for this omission. We initially used a standard alpha of .05 without correcting for the number of tests. However, we realised that this approach might be overly lenient and may not guard against incidental results like the ones noted by the Reviewer in the highlighted sentence. We now include a correction by adjusting the alpha via the Gao, Starmer, & Martin (2008) method. This approach uses a Bonferroni correction to adjust for the number of effective tests (M_{eff}). Following this approach, we have applied an experiment-wise alpha of .007. Changing the threshold for statistical significance had no impact on any main results or discussion. We have introduced the following paragraph in the Methods (lines 606-615):

Threshold for statistical significance (α value). Throughout the whole study, we used an adjusted alpha of $\alpha=.007$. The adjusted alpha was obtained via the Bonferroni correction:

$$\alpha = \frac{.05}{M_{eff}}$$

where $M_{eff}=7$, the number of effective tests, accounts for the dependency between variables. M_{eff} was obtained, following (Gao et al., 2008), as the number of eigenvalues that explained 99.5% of the variance across all the variables included in this study. (The correlation matrix included seven variables, i.e., the SMDT and BASrr total scores and the five BMRQ facets' scores; we have excluded the BMRQ total score as it is a linear combination of the five BMRQ facets' scores.) M_{eff} was computed using the `meff(method = "gao")` function from the `poolr` R package . Alternative methods (e.g., Li & Ji) resulted in a less conservative alpha (i.e., $M_{eff} < 7$).

Comment #18

Figure 3 depicts various results: the phenotypic correlations, the Cholesky decomposition results, and the implied decomposition of phenotypic variance. The parameter estimates of fig 3B are supposed to be given in Suppl Table 3. 1. The parameter estimates are not all visible in Fig 3B. So that is inconvenient. I suppose that the discrepancy between the Fig 3 b results and the supply Table 3 results is explained somewhere.

We are grateful to the Reviewer for noticing this. The revised Supplementary Table 5 includes all coefficients and additional statistics. The inconsistencies in Fig. 3 (now Fig. 2) were due to a few transcription errors. All typos have been corrected. Note that typos were only present in panel **b** (now panel **a**) and did not impact other panels. For completeness, we refer to Supplementary Data to reproduce Figures in our GitHub repo.

Estimates for Fig. 2 can be found following the link below:

https://github.com/giacomobignardi/h2_BMRQ/blob/main/SFILE/03_supplementary_data_2.csv

Comment #19

Figure 4 depicts various hypotheses concerning the relationship between facets... Are these the 5 facets of the BMRQ (I assume, this could be stated explicitly). The representation as path diagrams is hard to follow. 4A why is the orange colored factor call a common factor given that it only varies on 1 on the 5 facets / phenotypes? What does the "=" signify in 4A? I assume that 4B is a correlated factors model. The pathdiagrammatic representation is not correct (given path diagram conventions). Is the about the 5 BMRQ facet? Why are there 4 variables? It is strange that in 4A the hypothesis of uncorrelatedness extends to the E influences. Why would the hypothesis concerning A (Genetic) influences extend to the E influences (in 4A and 4B, but not in 4C). The explanation of 4C is hard to follow: "The common latent genetic factor

(in orange) could 261 explain all the genetic variance associated with one facet (e.g., dashed circle).” The dashed circle does not pertain to the all the genetic variance, it pertains to the residual variance. If the dashed latent variable has zero variance, the correlation between the associated phenotype and the common factor is 1. That would imply that the common factor and the observed variable are indistinguishable (meaning that we do not really need the common factor at all). All this is about modeling the associations between the 5 facets, but are the facet correlations actually reported? Why is reference to these “Phenotypic correlations can be found in Supplementary Fig. 4” in the figure caption of fig 5. The correlations are the focus, but are hardly mentioned.

We are glad to have had the opportunity to modify Fig. 4 (now Fig. 3) to enhance understanding. In summary, we have removed panel a and introduced a new panel c to extend the representation to the environmental component. We have made this more apparent in the Fig. 3 caption and refer to the Supplementary Material for the more appropriate path-diagrammatic representation.

We also thank the Reviewer for noting that more emphasis was needed on the phenotypic correlations. We have included a table (Table 2) with the phenotypic and within- and between-trait twin correlations.

Comment #20

L 234 The results concerning the 5 facets are based on an AE model. Are the twin correlations reported? Somewhere in the suppl perhaps. This twin correlations are important info, really. Note that the choice for an AE model implies that an ACE or ADE is not suitable? Was that established?

We thank the Reviewer for this suggestion. We now have added implied twin correlations for the facets in Table 2.

Further, we have carried out and reported model comparison tests between ADE or ACE (the latter expected only for the social reward facet) versus AE models. Model comparisons indicate that dropping the D component from the facets did not substantially worsen the model fit for the AE models. Results can be found in Supplementary Table 6. Regardless, the (non-significant) non-additive genetic components did vary substantially in size for the different facets, and we therefore now added to the discussion that, if present but not detected because of inadequate power, non-additive genetics effects for some facets would have overestimated the A component (lines 504-507).

Further, although we did not detect significant non-additive (i.e., dominance) genetic effects on music reward sensitivity and its facets, our estimates for twin-based heritability may

represent broad-sense heritability as any non-additive genetic variance would be pushed into the estimate for the variance of the A component.

Comment #21

Results in Fig 5 are given in terms of h^2 and A and E correlations. The reader may ask: how much does A contribute to the phenotypic covariance of (say) emotion evocation and mood reg? That question is not simple to answer based on this info. A genetic correlation is obviously informative (e.g. pleiotopy) as is h^2 (individual differences / variance), but is not the question here: what do A and E contribute to the phenotypic covariance? The anders, I think, is 50% vs 50%, notwithstanding the A correlation of .74.

The reviewer is correct, and the answer to their question is indeed ~50%, notwithstanding a r_A correlation of .74. We have now augmented our analyses to include bivariate heritability estimates (and equivalents for the environmental component) between facets. Results can be found in Supplementary Fig. 9.

Comment #22

The path diagram 6A does not include correlated residuals among the dependent variables (5 facets). Is that correct? Does this path diagram correspond to the hypotheses of interest? Namely uncorrelated residuals? Why is it that now the E model mirrors the A model (in contrast to the previous). Why does 6A include the unit vector (triangle) where as most of the other path diagrams do not? Note that in the text various statistical test outcomes are mentioned. As usual here without explicit reference to alpha or alpha given multiple testing (beyond the $<.05$, which suggests that the alpha is .05 throughout as applied to every single statistical test ... how many tests? What is the policy w.r.t. multiple testing).

The actual model does include all the correlated residuals. We have also removed panel a and opted for a verbal description instead. Since we have now broadened the scope to environmental sources of variance (and covariance) in music reward sensitivity facets, Fig. 6 (now Fig. 5) also highlights environmental correlations. Regarding multiple test corrections, as discussed above, our alpha is now .007. Importantly, this did not impact the main results.

Comment #23

318 Given the lack of genetic dimensionality (5 facets do not fit a common factor model), this must mean that the phenotypic structure is not unidimensional (even though the the E structure could be unidimensional – although this is apparently not of interest here). So, what light does that shed on the analyses of the facet total score, and on the results: L 124 “A confirmatory factor model showed acceptable fit for a model with a single latent music 125

reward sensitivity factor capturing correlations between the five facets (CFI = .96, SRMR = 126 .035).” And on the “notably high heritability” of the BMRQ total score? The authors note correctly: “Our results thus may challenge the epistemological status of music reward sensitivity as a latent causal factor 43,55,56, as a latent factor is unlikely to hold unless a common-genetic factor solution holds (for additional conditions, see 43).” Here again the focus on A structure. I suppose that the additional conditions concern the E structure? And the common pathway model. Again, it is unclear why the E structure (and E in general) is discarded.

Please note that we have now extended analyses to the E structure (results also do not support a shared environmental factor across facets of music reward sensitivity). As noted in the discussion, our result implies that genetic and environmental effects of music reward sensitivity are partly heterogeneous. We have clarified this in the manuscript (lines 406-410).

Here, we did not find support for a single overarching shared genetic factor of music reward sensitivity, suggesting no single common intermediate pathway from genes to different aspects of music enjoyment (with pathway, we refer to the combination of processes along the complex chain from genetic to phenotypic differences as in ref [see the main text for reference]). On the contrary, we found partly distinct and partly overlapping genetic pathways to music enjoyment.

We also clarified that we have an extensive discussion in the Supplement (Supplementary Note 4) regarding the impact of this expected genetic heterogeneity on brain-behaviour studies of music reward sensitivity (both past and present). Once more, we had to relegate this nuanced discussion to the supporting information due to the length of the current version of the main text.

Finally, we note that, as discussed below, we have opted to substitute “notably high” with a factual description of the estimate (i.e., 54%) and refrained from further qualitative descriptions of our results.

Comment #24

319 “Instead, these findings are consistent with musical enjoyment being built upon genetically interconnected yet partly distinct parts.”

This should be “individual differences in musical enjoyment”

Corrected.

Comment #25

326 *“Yet, the finding of notably high heritability for music*
327 *reward sensitivity gives hope for molecular genetic studies to answer questions about*
328 *genetic underpinnings of musicality in general and musical affect in particular.”*

Notably high heritability is .54 (in the AE model)? Why is that “notably” high? Apparent the e^2 ($\text{var}(E) / \text{var}(ph)$) is also “notably high” (.46)? I suppose that .54 would be notably high if we had expected low h^2 , but no such expectation was expressed

“Notably high” was initially referring to the heritability of other comparable traits, e.g., general reward sensitivity (e.g., 33% as in this study), and sensitivity to other forms of arts (e.g., chills from visual art and poetry, 36%, aesthetic evaluation, 26% to 41%, Bignardi et al., (2022, 2024), respectively). However, we agree with the referee and now simply provide the actual heritability estimate.

Comment #26

Whence this emphasis on passive gene-environment correlation? The participants are adults, the absence of C in adults does not imply an absence of C in children, where passive gene-env correlation may be relevant. The discussion is about passive $\text{cov}(AC)$. Why would that apply in adults? Btw: musical talent, and the positive feedback loop between genetic disposition and env circumstances is a primary example of active A-E covariances due to niche-picking (but this may apply to music production in proficient musicians).

We agree with the reviewer that in an adult sample, passive rGE (which will be captured in a C component) is of less of interest than active rGE (which will be captured in the A component). We have therefore added the following sentence (lines 492-503):

The lack of common environmental effects in adulthood, as found in this study, does not imply a lack of common environmental effects on similar musicality traits earlier in the lifespan. On the contrary, common environmental effects seem to contribute to variability in musicality traits in younger children, implying possible passive gene-by-common environment correlation (e.g., parents may both pass genetic variants and provide musically enriched environment contributing to music reward sensitivity of their children) or interactions (e.g., additive genetic effects associated with music reward sensitivity might vary within different musically enriched environments that the family provide to the children) at earlier stages in life. As such, an important venue for future research will be to quantify whether gene-environment correlations or interactions are at play in music reward sensitivity at earlier stages in life and how much they impact quantitative genetic estimates for musicality traits more broadly.

Regarding active gene-environment correlations, we are aware that music is a primary example of niche-picking in classic behaviour genetics textbooks (e.g., Knopik, 2015). Indeed, we have discussed active gene-environment correlations as an exciting avenue to pursue (from line 483).

Comment #27

417 "An additional assumption is the lack of gene-by-shared environment interaction,
418 which could lead to an underestimation of the variance of the C component." AxC
unmodelled results in overestimation of A variance? See L 562 "If AxC is present, then σ^2
% 563 " > σ^2 is expected"

We have amended the sentence in the discussion to reflect what we have already stated in the Method section.

Reviewer #2 (Remarks to the Author):

This was a very interesting article reporting on a genetic decomposition of music reward and its association with music perception and general reward sensitivity. The article is important as there have not been other genetic investigations on music reward sensitivity and how they relate to other constructs. I found the manuscript very well-written and appreciated the authors' succinct notes on where model assumptions were made and what biases would be introduced if these assumptions were violated. I therefore struggled to find places to give critical feedback. Below I note some comments, but please note that I consider almost all these comments minor. My most substantial concern is that I think some parallel questions/analyses should address environmental influences in addition to genetic influences (see below).

We thank Reviewer 2 for their comments. Below, we highlight how we addressed all the important points raised.

Comment #1

Introduction:

This section was well-written and clear. My only concern is that the article focuses primarily on genetic influences, but the nature of the twin dataset is also informative to the environmental influences on music reward and its overlap with other traits. I think unpacking these genetic associations is very interesting and important, but I suggest expanding the key research questions (e.g., lines 97-101) to include genetic and environmental variation, as these are two sides of the same coin.

Similar points were raised by Reviewer 1. We have now added analyses of the environmental influences to the manuscript. Specifically, we have added this paragraph in the introduction (lines 125-127):

All analyses were complemented by an investigation of environmental influences on music reward sensitivity and their overlap with environmental effects on music perceptual ability and general reward sensitivity.

Comment #2

Results:

It's unclear why the authors state that comparison of within-pair MZ and DZ correlations are an estimate of narrow-sense heritability. I would argue that this is an estimate of broad sense heritability, especially since they state that $r_{MZ} > r_{DZ}$ which indicates evidence for both additive and nonadditive genetic components (i.e., broad-sense heritability).

We apologise for the confusion and thank the Reviewer for this comment. We have corrected the statement accordingly.

Comment #3

I like that Figure 1 displays both the results for the ADE and AE solutions of the initial model. However, I would use the phrase 'more parsimonious' rather than 'better fit' when referring to the selection of the AE model over the ADE model. The AE model did not fit significantly worse than the ADE model, but it also does not fit better (e.g., it has the same AIC and BIC values), it is just a more parsimonious model.

We are thankful for this spot-on comment. We have amended the previous sentence to reflect the Reviewer's comment.

Comment #4

Either in the Figure 2 caption or in lines 187-191 it would be good to mention that these three possible outcomes could be observed for environmental influences as well. I don't think there is reason to expect environmental influences will only be 'partial separation' as depicted (Note: it makes sense for the figure itself to emphasize the genetic differences, I just suggest adding a note that these possibilities are displayed for genetic influences but apply to environmental influences as well).

As mentioned above, we have now complemented our manuscript with environmental variance/covariances analyses. We note, however, that we have now removed Fig. 2 as we realised it did not help in making the analyses more clear.

Comment #5

I can be convinced to keep Figure 2B as is, but it may help readers unfamiliar with genetic analyses to present a standardized version of this figure in the main text. For example, it makes sense that the main text and Figure 2C decompose variance in music reward sensitivity, but it's hard to tell what the heritability and genetic/environmental covariance for music perceptual abilities and general reward sensitivity are from this figure without doing a lot of computations by hand.

We agree with the Reviewer that unstandardised estimates may be daunting for the unfamiliar reader. We have now complemented Fig. 2a with standardised estimates obtained from a Cholesky decomposition. For simplicity, we have put the squared root of the path's estimates (and made this explicit in Fig. 2 caption) so that the reader can simply read the percentage of variance explained. We further complemented Fig. 2b by adding the variance components for music perceptual abilities and general reward sensitivity.

Comment #6

Given the emphasis on model selection in Figure 2, it would help to show specific tests to determine that the trivariate genetic model supports partial separation vs. complete separation of genetic or environmental influences on music reward vs. other phenotypes (e.g., chi-square test when genetic/environmental cross-paths are removed). Again, I think this is important to report for both genetic and environmental influences.

We have now added a model comparison to test the significance of each genetic and environmental path (from line 224).

Removing the genetic cross-paths from music perceptual abilities and general reward sensitivity to music reward sensitivity, respectively, worsened the model fit ($\chi^2(1)_{\Delta df} = 94.37$ and $\chi^2(1)_{\Delta df} = 87.71$, both $p < .001$), indicating partial overlap between genetic factors ... Environmental analyses revealed that only removing the environmental cross-path from general reward sensitivity to music reward sensitivity worsened the fit of the Cholesky ($\chi^2(1)_{\Delta df} = 64.07$, $p < .001$), while removing the one from music perceptual abilities did not ($\chi^2(1)_{\Delta df} = 0.004$, $p = .95$). This indicates a small overlap in environmental influences between general reward sensitivity—but not music perceptual ability—and music reward sensitivity, which ...

Comment #7

In lines 264-271, please report whether the environmental influences also support a distinct factor solution (like genetic influences). Also, it was hard to tell from Supplementary Table 4 whether distinct vs. common factors were tested for genetic and environmental influences separately (which it probably should be), or if the common factor model that was rejected collapsed both A and E influences at the same time.

We have supplemented our main analysis by implementing a hybrid-independent environmental pathway model. Results support partly distinct environmental paths to music enjoyment, effectively mirroring the findings at the genetic level (from line 290). We have added these findings to the discussion (see below). Note that the distinct vs. shared factor solutions were tested for genetic and environmental influences separately.

Comment #8

Discussion:

If you agree with my earlier suggestions, you will need to comment on whether the nonshared environmental influences show similar patterns as the genetic results in the discussion. Even if they don't, these findings would still be relevant to our understanding of music reward and related constructs. Note: the lack of shared environmental influences is addressed in a compelling way here, but I think some discussion should also talk about nonshared environmental influences as well.

We have added the following paragraph to comment on the novel findings on unique environmental effects (from line 444).

We also show that environmental influences broadly mirror genetic influences on music reward sensitivity and its facets. Although effects explained a similar magnitude of the variance in music reward sensitivity facets (48% to 58%), we also found no support for a single shared environmental factor common across facets of musical enjoyment. Yet, compared with genetic overlap, we found notable exceptions: First, environmental effects on music perceptual abilities and music reward sensitivity were non-overlapping. Second, environmental correlations between general and music reward sensitivity were similar. Although it would be tempting to speculate on why this is the case, it is challenging to interpret environmental effects and correlations as the environmental component includes all residual sources of variability left after accounting for genetic effects, including measurement error. Furthermore, music perceptual abilities and general reward sensitivity were collected at different time points, possibly attenuating correlations between the two traits and making it more difficult to draw inferences on differences in non-common environmental effects.

Comment #9

It would be useful to comment briefly in the limitations paragraph on how the 10 year gap between administration of the SMDT and reward measures may impact the results. I suspect this would underestimate phenotypic and genetic/environmental correlations with SMDT.

As pointed out by the reviewer, the time gap may have led to an underestimation of the phenotypic correlations between the SMDT and the BMRQ. We have highlighted the gap and possible attenuation (see response above): “...Furthermore, music perceptual abilities and general reward sensitivity were collected at different time points, possibly attenuating correlations between the two traits and making it more difficult to draw inferences on differences in unique environmental effects.”

Reviewer #3 (Remarks to the Author):

I appreciated reading this interesting study, which uses behavioural data from a large sample of Swedish twins to investigate genetic factors underlying music enjoyment. On a general level, its interdisciplinarity, breadth, rigour, and potential scientific and popular impact make it appear well-suited for eventual publication in Nature Communications. I do not feel confident enough in the specific details of the methods to make strong comments about them, but from what I can tell they mostly seem to be reasonable methods and their conclusions largely seem reasonable. So I mainly only have relatively minor suggestions that I think might help clarify the study and better highlight its strengths and limitations:

We thank the Reviewer for their very positive and constructive minor suggestions. Please see below for our point-by-point response.

Comment #1

Restricted sample: The main limitation I see is that the sample is not only restricted to twins, but SWEDISH twins. I was expecting to see some discussion of this after the discussion on limitations of twin studies, so was surprised to see it not mentioned at all, as if we could assume that genetic and environmental factors underlying musical reward in Swedish twins could be assumed to generalize to all humans throughout the world. The issue of non-representativeness of samples from Europe and North America is increasingly recognised as one of the key challenges of music cognition (Jacoby et al., 2020) and the behavioural sciences more generally (Henrich et al., 2010), so certainly seems to deserve more acknowledgment.

We are thankful to the reviewer for this excellent suggestion. We have expanded our discussion to acknowledge that results were obtained on a European sample and caution against generalizability. The following section can be found in text, lines 514-521:

... although a recent study in Norwegian twins reported a similar heritability estimate for sensitivity to music, we caution against a generalisation of our results to non-Northern European populations. It is important to keep in mind that heritability estimates may be both population- and environment-specific (for misconceptions about heritability, see ref in the main text). Here we have focused our analysis on a particular musicality trait shown to be present in diverse populations. Whether the genetics of individual differences in music enjoyment are similar across populations is yet to be studied.

Comment #2

Title: I find the title, “Distinct genetic pathways to music enjoyment” overly vague. Best practice would be to give at least a hint of the data/methods (<https://www.nature.com/articles/s41562-023-01596-8>)(<https://www.nature.com/articles/s41562-023-01596-8>). In particular, I think you really should include “Swedish twins” somewhere in the title (see above), and perhaps an indication of the BMRQ or more generally that you are analysing behavioral data. For example, “Behavioral data from Swedish twins reveals distinct genetic pathways to music enjoyment”.

We thank the Reviewer for the suggestion and the links to best practice. We implemented the Reviewer’s recommendation to include “twin” in the title:

Twin Modelling Reveals Partly Distinct Genetic Pathways to Music Enjoyment

However, in line with the guidelines provided (<https://www.nature.com/articles/s41562-023-01596-8>), we prefer to mention the studied population in the abstract rather than in the title, since our results are not “unlikely to be relevant for understanding other populations”. On the contrary, we believe that our study, by showing implied genetic associations within a population, opens a window into novel research that could test whether similar genetic mechanisms are detected across populations and, as such, are extremely relevant. Including “Swedish” in the title could give the unfounded impression that we have evidence that the findings are specific to the Swedish twins, which we obviously do not. We have also added a discussion of this topic to the limitations.

Comment #3

I should mention here that the term “pathway” also feels a bit vague - to me it implies a clearer mechanism linking genotype and phenotype than I see here. However I’m not confident I can necessarily offer a better term - “factor” came to mind, but I’m not sure it’s

quite what is needed. In any case, I suggest considering the term and perhaps defining it more clearly if you intend to continue using it so prominently (and maybe also avoiding using confusingly similar terms like “the auditory cortex and its pathways...”)

With *pathway*, we refer to the combination of processes along the complex chain from genetic to phenotypic differences (Clapp Sullivan et al., 2024). We have now expanded our discussion to make this more clear (lines 406-410).

Here, we did not find support for a single overarching shared genetic factor of music reward sensitivity, suggesting no single common intermediate pathway from genes to different aspects of music enjoyment (with *pathway*, we refer to the combination of processes along the complex chain from genetic to phenotypic differences as in ref [see ref in the main text]). On the contrary, we found partly distinct and partly overlapping genetic pathways to music enjoyment.

Abstract:

“genetic factors substantially explain variance in music reward sensitivity above and beyond genetic influences shared with music perception and general reward sensitivity” -I think this would benefit from a quantitative indication here - for example, the relative size of genetic and environmental effects?

We agree the text would benefit from some quantitative take-away. We have updated the abstract sentence as follows.

We estimate that genetic effects contribute up to 54% of the variability in music reward sensitivity, with 70% of these effects going above and beyond music perception and general reward sensitivity.

Comment #4

“large sample of Swedish twins (N = 9,169)” - this is good to mention this here, but isn’t “N = 9,169” a little misleading if this only included 2305 pairs as suggested in Table 1? Would something like “N = 4,610 individuals (2,305 pairs)” be more accurate? Apologies if I’ve misunderstood the methods.

We provide more details to clarify this in the Abstract.

Comment #5

“behavioural data on several facets of music reward sensitivity, music perceptual ability, and

general reward sensitivity” - might be good to specify that the “primary measure” is the BMRQ.

We thank the reviewer for this suggestion. We have adapted the abstract as suggested.

Comment #6

If you have room, I suggest it might be worth mentioning the intriguing result that “music perception shows stronger genetic correlations with social bonding than other facets of music reward” as you discuss later in the paper.

After implementing all previous suggestions, we are afraid we cannot find additional space to highlight this finding in the abstract.

Comment #7

Novelty of data:

It wasn't obvious to me how much of the analyses reported analysed new data and how much was reanalysis of existing data. For example: ““a large sample of deeply phenotyped monozygotic (MZ) and dizygotic (DZ) twins with available musicality data” - where do this “available musicality data” and “large sample” come from? Ref. 13 is cited at one point but it's not immediately clear to me how much (if any) of the data reported was previously published in ref. 13. I think it should be clear to a reader of this article what is new data and what is previously published data without having to read external references. This is particularly important since the current data are not made publicly available (“as registry data were used” , which sounds reasonable), and so cannot be independently checked.

The data all come from the Swedish Twin Adults: Genes and Environment (STAGE) cohort of the Swedish Twin Registry. In 2011, some of the authors of this study initiated the first and most extensive twin study on music-related behavior. The Swedish Twin Music Discrimination Test (SMDT, used here to measure music perceptual abilities) was administered during this first wave of data collection and later validated (Ullén et al., 2014). In 2022, a second wave of data collection was conducted, including questionnaires like the Barcelona Music Reward Questionnaire and the Behavioral Approach System. The data obtained from this latter wave (i.e., BMRQ and BAS) are novel and have not been analysed before. We have updated the method section to clarify differences in waves of data collection (see below; lines 544-547), and made this more evident in Table 1.

In the first wave of data collection, participants completed the Swedish Musical Discrimination Test (see below). More details on this wave can be found in Ullén et al..

Additionally, in the second wave, the Behavioral Approach System and Barcelona Music Reward Questionnaire were administered.

Comments #8, #9, and #10

Figures:

In general, I found the figures a bit confusing. The main problem is that they were peppered with acronyms that were difficult to interpret (e.g., “E mrs”, “DZos”, “ee”, etc.). I suggest trying to re-draw some to write out key acronyms in full in some places, remove them if not essential, and otherwise try to at least define them more clearly in the caption.

In particular, the letters “A” and “E” appear to have much importance, but I can’t tell what they represent. Does “A” mean “additive genetic” and “E” mean “environmental” (inferred from the Fig. 5 caption)? If so please make that clearer!

3b: white text boxes seem to be obscuring some lines

We have adapted all five figures to accommodate all Reviewers' requests, increase clarity, and reduce jargon as much as possible.

Comment #11

Discussion:

Given the close relationships between music and language (e.g., Nayak et al., 2022; Patel, 2008), I would have liked to see some speculation about whether we expect the “pathways” to be distinct or shared with language (e.g., reward from conversation).

We find this question on the overlap between genetic pathways between language and music exciting (some of the authors have spent some considerable time in the past tackling it from an empirical stance in other studies, e.g., Alagöz et al. (2024) and Wesseldijk et al., (2023)).

However, since we are unaware of literature on specific sensitivity to rewards from language (e.g., reward from conversation), we prefer to refrain from speculating too much on this. It represents an interesting topic for future studies.

Comment #12

“genetic variance associated with music reward sensitivity, beyond perceptual and general reward processing” - what is the mechanism here?

This is a question that, at the moment, cannot yet be answered based on the available data. However, in the discussion, we noted that genetic influences on music reward sensitivity, beyond perceptual and general reward processing, “mirrors the finding that specific musical anhedonia, i.e., blunted or absent hedonic responses from music stimuli, exists in the absence of any perceptual or generalised reward deficit.” We have speculated, both in the discussion and more in-depth in Supplementary Note 4, that such genetic variation may underpin individuals’ variability in both functional and structural connections between auditory and reward systems, the same connections that are altered in cases of congenital musical anhedonia.

Comment #13

“different aspects of musicality, namely producing music and enjoying music, might follow different patterns of intergenerational transmission” - seems to imply that “intergenerational transmission” of music(ality) is only via genes but of course cultural transmission is probably equally or more important. Better to use different words like “genetic inheritance” here if that’s what you mean.

Intergenerational transmission is a standard term for the process of phenotypic transmission between parents and offspring, including genetic (e.g., additive genetic) and non-genetic (e.g., cultural transmission) processes and their interaction (Branje et al., 2020).

Comment #14

“Anirudh” in Acknowledgments missing an extra “d”

Ops. We thank the Reviewer very much for pointing out this typo and have corrected the error.

Comment #15

“Exploratory analyses” - typically refer to non-preregisterd analyses, contrasted with “confirmatory” analyses that are pre-registered. But here no analyses are pre-registered, are they? Regardless, I think Nature Communications requires statements about whether analyses are pre-registered, which I don’t see in the current manuscript.

The study was not pre-registered. We have now amended the relevant section and included a statement in the method section (line 803).

References

- Alagöz, G., Eising, E., Mekki, Y., Bignardi, G., Fontanillas, P., Nivard, M. G., Luciano, M., Cox, N. J., Fisher, S. E., & Gordon, R. L. (2024). The shared genetic architecture and evolution of human language and musical rhythm. *Nature Human Behaviour*, 1–15. <https://doi.org/10.1038/s41562-024-02051-y>
- Bignardi, G., Chamberlain, R., Kevenaar, S. T., Tamimy, Z., & Boomsma, D. I. (2022). On the etiology of aesthetic chills: A behavioral genetic study. *Scientific Reports*, 12(1), Article 1. <https://doi.org/10.1038/s41598-022-07161-z>
- Bignardi, G., Smit, D. J. A., Vessel, E. A., Trupp, M. D., Ticini, L. F., Fisher, S. E., & Polderman, T. J. C. (2024). Genetic effects on variability in visual aesthetic evaluations are partially shared across visual domains. *Communications Biology*, 7(1), 1–15. <https://doi.org/10.1038/s42003-023-05710-4>
- Branje, S., Geeraerts, S., de Zeeuw, E. L., Oerlemans, A. M., Koopman-Verhoeff, M. E., Schulz, S., Nelemans, S., Meeus, W., Hartman, C. A., Hillegers, M. H. J., Oldehinkel, A. J., & Boomsma, D. I. (2020). Intergenerational transmission: Theoretical and methodological issues and an introduction to four Dutch cohorts. *Developmental Cognitive Neuroscience*, 45, 100835. <https://doi.org/10.1016/j.dcn.2020.100835>
- Cinar, O., cre, & Viechtbauer, W. (2022). *poolr: Methods for Pooling P-Values from (Dependent) Tests* (Version 1.1-1) [Computer software]. <https://cran.r-project.org/web/packages/poolr/index.html>
- Clapp Sullivan, M. L., Schwaba, T., Harden, K. P., Grotzinger, A. D., Nivard, M. G., & Tucker-Drob, E. M. (2024). Beyond the factor indeterminacy problem using genome-wide association data. *Nature Human Behaviour*, 1–14. <https://doi.org/10.1038/s41562-023-01789-1>
- Gao, X., Starmer, J., & Martin, E. R. (2008). A multiple testing correction method for genetic association studies using correlated single nucleotide polymorphisms. *Genetic Epidemiology*, 32(4), 361–369. <https://doi.org/10.1002/gepi.20310>
- Neale, M. C., & Eaves, L. J. (1993). Estimating and controlling for the effects of volunteer bias with pairs of relatives. *Behavior Genetics*, 23(3), 271–277. <https://doi.org/10.1007/BF01082466>
- Wesseldijk, L. W., Gordon, R. L., Mosing, M. A., & Ullén, F. (2023). Music and verbal ability—A twin study of genetic and environmental associations. *Psychology of Aesthetics, Creativity, and the Arts*, 17(6), 675–681. <https://doi.org/10.1037/aca0000401>

We are grateful for the final suggestions from all referees. We have made minor adjustments based on the comments and responded to each below. Please see the original suggestions *in italics*, followed by our response in blue. Textual changes are reported in red.

Giacomo Bignardi,
On behalf of the Coauthors.

REVIEWERS' COMMENTS

Reviewer #1 (Remarks to the Author)

Twin modelling reveals partly distinct genetic pathways to music enjoyment (revision).
The authors have acted adequately on all my previous remarks.

Many thanks!

Here are some residual remarks.

Figure 1 C header is incorrect. These are not DZ twins, they are MZ twins.

Figure 1C header is correct as these may represent both MZ, DZ, and DZ opposite-sex twins. The expected additive (r_α) and dominance (r_δ) genetic correlation are set to .5 and .25, respectively, for DZ (so the reported estimates for the covariances between the A and the D latent factors for DZ are respectively $\frac{1}{2}$ and $\frac{1}{4}$ of the MZ ones). Details are in Fig. 1C caption.

*“Footnote: Circles are the latent component;199 Double-headed arrows within circles,...”
“Circles are the latent component”. Does not make sense because component is singular.
Also the term component here is ambiguous: Circles represent latent variables.
The double headed arrow is not “within circles”, Some connect different variable, other connect the variable with itself. Within the circle implies that the double headed arrow is *within* the circle.*

We have changed the Figure caption as suggested.

I do not understand the notation “Squares|Rectangles”

The notation stands for “or”, i.e. squares or rectangles. For simplicity, we have amended the sentence.

“the triangle, the phenotypic mean grouped by twin order201 (panel a) and sex (panel c) already adjusted for age;”

The triangle represents the unit vector not the phenotypic mean. The regression of the phenotype on the unit vector is standard in linear regression to estimate the intercept, which

coincides with the phenotype mean here. “by twin order” refers to what ordering? how are they ordered.

We have made this distinction clear in Fig. 1 caption and amended inconsistencies. The twin order is randomised within pairs, but for opposite-sex dizygotic twins, for which women are always first.

223. “Removing the genetic cross-paths from music²²³ perceptual abilities and general reward sensitivity to music reward sensitivity, respectively,²²⁴ worsened the model fit ($\chi^2(1)\Delta df = 94.37$ and $\chi^2(1)\Delta df = 87.71$, both $p < .001$), indicating partial²²⁵ overlap between genetic factors (see Supplementary Table 5).”

Given the fact that the readership is not necessarily familiar with the Cholesky parameterization of a covariance matrix, but would be useful to add a sentence. E.g., concerning $\chi^2(1)\Delta df = 94.37$. What is this a test of, not formulated in terms of the Cholesky. E.g., say, “This is a test of the hypothesis that the correlation between the additive genetic variable of music perc abilities and the additive genetic variable of music reward sensitivity is zero”.

We have added a sentence to explain better the rationale behind the model comparison

To test whether the three phenotypes displayed partial overlap between genetic factors, we compare the full Cholesky decomposition with models in which genetic cross-paths from music perceptual abilities or general reward sensitivity to music reward sensitivity were set to zero, respectively.

Btw: the notation is weird. Should this be $\chi^2(\Delta df=1) = 94.37$?

Good point! We have adapted the suggested notation throughout the manuscript.

I assume that the authors like the Cholesky because of the “sequential nature” (in the linear regression sense), but ultimately the representation of the completely standardized AE model with correlations among the three additive genetic variables and the three E variables, and $\sqrt{h^2}$ and $\sqrt{1-h^2}$ or $\sqrt{e^2}$ is easier to understand.

Correct! The sequential nature of the Cholesky allowed us to estimate genetic effects on music reward sensitivity over and above the influences of music perceptual abilities and general reward sensitivity. Thus, the Cholesky provided the most appropriate modelling solution to answer our second research question: “To what extent do genetic effects influence music reward sensitivity above and beyond genetic effects shared with music perceptual abilities and general reward sensitivity?”. Therefore, we believe the Cholesky and the associated Figure 2 best represent our findings.

Figure 4 “b Genetic and environmental effects³¹⁵ on music reward sensitivity are partially heterogeneous.” What does heterogeneous mean here?

It means that the genetic effects of one facet (e.g., emotion evocation) are not fully shared with another facet (e.g., sensory-motor). Regardless, we have removed this sentence from

the Fig. 4 caption, as this is better explained later in the text.

“The likely absence⁴⁶⁴ of shared environmental effects may imply only a small, if present, passive gene-environment⁴⁶⁵ correlation (e.g., genotypes associated with music reward sensitivity in the parents influence⁴⁶⁶ the children via the environment the parents provide and the genes they pass on to their⁴⁶⁷ children, see ref.60).”

The result was obtained in adult. The absence of C does not imply anything about cov(AC) in children. E.g., there is influence of C and covAC in children’s IQ. But there is not C in adults IQ. How do the results obtained in adults generalize to possible interplay in children? See “The lack of common⁴⁹¹

environmental effects in adulthood, as found in this study, does not imply a lack of common⁴⁹²

environmental effects on similar musicality traits earlier in the lifespan.”

Passive gene-environment correlations refer to indirect genetic effects which are environmentally mediated. The paragraph on passive gene-environment correlations was meant to underscore that, if present, passive gene-environment correlation acts as a common environment when using the classical twin design. Since we detected no common environment, we simply stated that this suggested no passive gene-environment effects on variation in music reward sensitivity in adulthood. This is relevant for future research, as passive gene-environment correlation is known to complicate efforts for detection of genetic associations (e.g., SNP-based heritability). Assuming that the effects of passive gene-environment effects on individual differences may become less important with increased age, as it is suggested for other complex traits, the fact that we find little evidence for passive gene-environment correlations in a middle-aged sample is, as such, not surprising and we agree that it holds little, if any, information on whether passive gene-environment correlation may play a role in reward sensitivity in childhood. To avoid further confusion, we have adapted part of the paragraph on passive gene-environment effects.

“On the contrary, this is to be expected if evocative and active gene-environment⁴⁸⁴ correlations are at play, which seems likely for traits related to music enjoyment. Such gene-⁴⁸⁵ environment correlations would not inflate heritability estimates.”

*But Purcell (2002 Twin Research Volume 5 Number 6 pp. 554± 571) demonstrates that
AxE inflates E variance in the CTD
AxC inflates A variance in the CTD
+rAC inflate C variance in the CTD
+rAE inflates A variance in the CTD*

So +rAC and +rAE inflate C and A variance respectively but the heritability (standardized A variance) is not inflated?

649 “The latter case would instead result in active gene-environment correlations that⁶⁵⁰ are still consistent with the estimate of the variance components.”

gene-environment correlations are consistent with the estimate of variance components.

This does not make sense. Also note that “the estimate” is singular and components is plural.(Remarks on code availability)

The Reviewer is correct: active gene-environment correlation will inflate estimates of additive genetic effects. We simply wanted to state that heritability estimates under active gene-environment correlation are consistent (within a causal framework) with additive genetic effects on the phenotypes (e.g., Morris et al. 2020). We have made some changes to make this less ambiguous.

Reviewer #2 (Remarks to the Author)

The authors have addressed all my concerns and they were very responsive to those concerns of the other reviewers. I believe this manuscript is suitable for publication(Remarks on code availability)

Many thanks!

Reviewer #3 (Remarks to the Author)

I thank the authors for their comprehensive response, which has resolved almost all of my concerns.

I'm still not 100% convinced by the justification of leaving out "Swedish" from the title, given the (good) new discussion about how we can't necessarily generalise these results outside of Northern Europe. However, I'm happy to leave the final decision on this to the authors/editors.

Many thanks. To avoid the unwanted impression of evidence of specificity to the Swedish population (e.g., by adding “in Sweden” at the end of the title), we have left “Swedish” in the abstract alone.

*Signed,
Patrick Savage(Remarks on code availability)*

References

Morris, T. T., Davies, N. M., Hemani, G. & Smith, G. D. Population phenomena inflate genetic associations of complex social traits. *Sci. Adv.* **6**, eaay0328 (2020)